# VEGF-B signaling impairs endothelial glucose transcytosis by decreasing membrane cholesterol content

Christine Moessinger, Ingrid Nilsson, Lars Muhl, Manuel Zeitelhofer, Benjamin Heller Sahlgren, Josefin Skogsberg & Ulf Eriksson[*] iD

## Abstract

Regulation of endothelial nutrient transport is poorly understood. Vascular endothelial growth factor B (VEGF-B) signaling in endothelial cells promotes uptake and transcytosis of fatty acids from the bloodstream to the underlying tissue, advancing pathological lipid accumulation and lipotoxicity in diabetic complications. Here, we demonstrate that VEGF-B limits endothelial glucose transport independent of fatty acid uptake. Specifically, VEGF-B signaling impairs recycling of low-density lipoprotein receptor (LDLR) to the plasma membrane, leading to reduced cholesterol uptake and membrane cholesterol loading. Reduced cholesterol levels in the membrane leads to a decrease in glucose transporter 1 (GLUT1)-dependent endothelial glucose uptake. Inhibiting VEGF-B *in vivo* reconstitutes membrane cholesterol levels and restores glucose uptake, which is of particular relevance for conditions involving insulin resistance and diabetic complications. In summary, our study reveals a mechanism whereby VEGF-B regulates endothelial nutrient uptake and highlights the impact of membrane cholesterol for regulation of endothelial glucose transport.

**Keywords** endothelial cell; glucose transcytosis; LDLR; membrane cholesterol; VEGF-B

**Subject Categories** Metabolism; Vascular Biology & Angiogenesis

## Introduction

The endothelium represents the first cell layer that nutrients have to pass on their way from the blood circulation to the underlying parenchyma. In most blood vessels, especially in capillaries, endothelial cells (ECs) are tightly connected forcing most nutrients to be transported through the cell body by specific transport systems.

Endothelial glucose uptake relies on specific transporters of the solute carrier (*SLC*) gene family. Particularly well characterized are the ECs of the blood–brain barrier, highly enriched with the *solute carrier family 2 member A1 (SLC2A1,* glucose transporter 1 (GLUT1)) [1,2]. Different transporters facilitate fatty acid (FA) uptake from blood, but the exact mechanisms that govern endothelial transcytosis of lipids is largely unknown [3]. However, in brain ECs, major facilitator superfamily domain-containing protein 2A (MFSD2A) has been identified as a lysolipid transporter governing brain uptake of the essential very-long chain FA docosahexaenoic acid [4,5].

In the blood, lipids either exist as free FAs bound to albumin or as lipoproteins. Lipoprotein particles consist of a core of neutral lipids, triacylglycerol (TAG) and cholesteryl esters (CE), surrounded by a monolayer of phospholipids harboring small amounts of non-esterified/membrane cholesterol and protein. Lipoproteins enter or cross the cell by receptor-mediated endocytosis or transcytosis. Endothelial uptake of low-density lipoprotein (LDL) is mediated by low-density lipoprotein receptor (LDLR) [6] or by scavenger receptor B1 (SRB1) [7], while high-density lipoprotein (HDL) transport relies on SRB1 and ABC transporter A1 (ABCA1) [8,9]. Inside the cell, LDL is degraded in the endosomal-lysosomal compartment. The released cholesterol is either converted to CE and stored in lipid droplets or integrated into cellular membranes. Non-esterified cholesterol controls membrane fluidity and integrity and can associate with other lipid species such as sphingolipids to form lipid rafts [10]. The association of particular membrane proteins to cholesterol-rich lipid rafts has been shown to be important for the regulation of their downstream signaling [11].

Though the majority of circulating nutrients will pass the endothelium, some are used by the ECs themselves. Glycolysis was recently identified as the main ATP-producing metabolic pathway active during highly metabolic angiogenic processes such as vessel sprouting [12]. Apart from utilizing glucose and glycolysis for energy production, proliferating ECs employ FAs and fatty acid oxidation as means of generating precursors of deoxyribonucleotide synthesis [13]. In contrast, quiescent ECs predominantly rely on fatty acid oxidation metabolism in order to support NADPH production aiming at sustaining redox homeostasis and counteracting oxidative stress [14].

Vascular Biology Division, Department of Medical Biochemistry and Biophysics, Karolinska Institute, Stockholm, Sweden
*Corresponding author. Tel: +46 8 52487109; E-mail: ulf.pe.eriksson@ki.se

FA and glucose metabolism are inversely linked with each other according to the Randle cycle theorem [15], which stipulates a reduction in glucose uptake and metabolism upon increased FA availability and *vice versa*. As implied by the Randle cycle, decreased glucose uptake in response to conditions inferring insulin resistance, *e.g.,* type 2 diabetes mellitus (T2DM) and the metabolic syndrome, are accordingly associated with increased deposition of circulating lipids in peripheral tissues [16–19].

We have previously reported that vascular endothelial growth factor (VEGF)-B signaling via VEGF receptor-1 (VEGFR1) and its co-receptor neuropilin-1 (NRP1) enhances endothelial FA uptake and transcytosis [20]. VEGF-B is widely expressed, *e.g.,* in myocytes of heart and skeletal muscle and in epithelial cells of the kidney tubular system [21]. In contrast to most other members of the VEGF family, VEGF-B exhibits only restricted angiogenic potential [22]. Overexpression of VEGF-B in the heart leads to increased growth of epicardial coronary vessels and cardiac hypertrophy, correlating with improved cellular survival and revascularization after myocardial infarction [23–26]. Mice lacking the *Vegfb* gene are reported to display a mild arterial conduction defect [27], or smaller hearts [28], respectively. In rodent models of dyslipidemia, pre-diabetes, or T2DM, deficiency or pharmacological inhibition of VEGF-B ameliorates ectopic lipid tissue accumulation, correlating with improved insulin sensitivity and glucose handling [17–19]. Similar effects could however not be reproduced in a smaller study performed on *Vegfb*-deficient mice [29], requiring a more detailed understanding of the function of VEGF-B signaling in ECs and impact on lipid and glucose metabolism.

Here, we show that VEGF-B signaling reduces the rate of endothelial glucose uptake, restricting tissue glucose availability. VEGF-B signaling impedes LDLR-mediated cholesterol uptake leading to reduced plasma membrane cholesterol loading, which negatively impacts GLUT1-facilitated glucose transport. We further demonstrate that ECs require a tightly controlled cholesterol metabolism in order to ensure optimal glucose supply to tissues. Notably, a reduced cholesterol uptake into ECs upon VEGF-B signaling correlated with decreased cholesterol content of the underlying tissue. Finally, tissue cholesterol homeostasis in mouse models of obesity and T2DM was restored in response to VEGF-B targeting, highlighting the possible impact of membrane cholesterol content on multiple signaling and transport systems involved in disease progression, and the involvement of VEGF-B signaling for its regulation.

## Results

### VEGF-B signaling reduces endothelial glucose transcytosis and cardiac glucose utilization

VEGF-B knockout ($Vegfb^{-/-}$) mice were shown to exhibit reduced lipid accumulation in heart and skeletal muscle, which was paralleled by increased cardiac glucose utilization as measured by 2-[$^{18}$F]fluoro-2-deoxy-D-glucose ($^{18}$F-DG) PET imaging [17,20]. We set out to explore whether VEGF-B affects glucose distribution directly or indirectly as a response to altered FA availability as implied by the Randle cycle theorem [15]. A 2-h stimulation of murine microvascular pancreatic ECs (MS1) and primary human

umbilical vein ECs (HUVEC) with the two different splice isoforms of VEGF-B, VEGF-B$_{167}$, or VEGF-B$_{186}$, led to reduced endothelial uptake of a fluorescent glucose analogue (2-NBD-glucose) (Fig 1A), along with increased FA uptake (Fig EV1A). These effects were seemingly independent of the origin of the vascular bed, since both heart and brain primary ECs responded similarly (Fig EV1B and C). In contrast to VEGF-B, VEGF-A did not affect EC glucose uptake capacity (Fig EV1D), suggesting a distinctive VEGF-B-specific signaling pathway. The impact of VEGF-B signaling on EC glucose uptake most probably reflected a difference in glucose uptake rate rather than glucose metabolism. This is supported by the observation that 2-NBD-glucose tracer net uptake did not reach saturation in HUVECs during a 40-min uptake session (Fig EV1E) [30,31]. A 2-h stimulation with VEGF-B also reduced endothelial glucose transcytosis, as evaluated using MS1 cells in a transwell 2-chamber system (Fig 1B). These results suggest that VEGF-B-regulated EC uptake and transcytosis may modulate physiological glucose delivery from the blood stream to the underlying tissues. Although ECs themselves use glucose for energy production [12], the majority is transported through the endothelium into, *e.g.,* skeletal muscle or heart, where it is used for ATP generation or stored as glycogen. We have previously shown that $Vegfb^{-/-}$ mice have increased glucose uptake in the heart [20] and could observe here that $Vegfb^{-/-}$ mice accordingly display increased cardiac glycogen content (Fig 1C). No difference in cardiac expression of genes involved in synthesis or degradation of glycogen was detected (Fig EV1F), suggesting a difference in glucose uptake rate, rather than changed partitioning of glucose or glycogen metabolism. In order to demonstrate whether VEGF-B influences cardiac glucose utilization directly *in vivo*, we applied VEGF-B-specific neutralizing antibodies (2H10) to acutely reduce VEGF-B signaling in high-fat diet (HFD) fed C57BL/6 mice, where the effect of VEGF-B signaling in metabolic regulation has been implicated previously [17]. Short-term (1 week) 2H10-treated HFD mice showed increased cardiac glucose accumulation during a 60-min $^{18}$F-DG PET acquisition compared to control antibody-treated mice (Fig 1D). The initial cardiac glucose uptake rate, represented as the slope of glucose uptake between 0.9 and 6.5 min, was higher in 2H10-treated mice (Fig 1E), pinpointing that the increased cardiac glucose accumulation in 2H10-treated mice is due to effects on glucose uptake rate and not on glucose metabolism. This is conceivably related to increased endothelial glucose uptake and transcytosis, in line with our *in vitro* observations.

### GLUT1 is vital for glucose transport in endothelial cells and implied in VEGF-B signaling

In muscle tissue, glucose uptake increases in response to insulin signaling. In contrast, stimulation of ECs (HUVECs) with a physiological insulin concentration did not evoke glucose uptake and only a minor increase was observed with a supra-physiological insulin concentration (300 nM, Appendix Fig S1A), which is in accordance with previous findings [32]. Insulin had moreover no effect on VEGF-B-mediated reduction in glucose uptake (Appendix Fig S1A). In line with these data, ECs isolated from mouse heart showed low expression of the insulin-sensitive *Slc2a4* glucose transporter (GLUT4) and sodium-dependent glucose transporters of the *Slc5a* (SGLT) family, whereas high expression levels for the insulin-

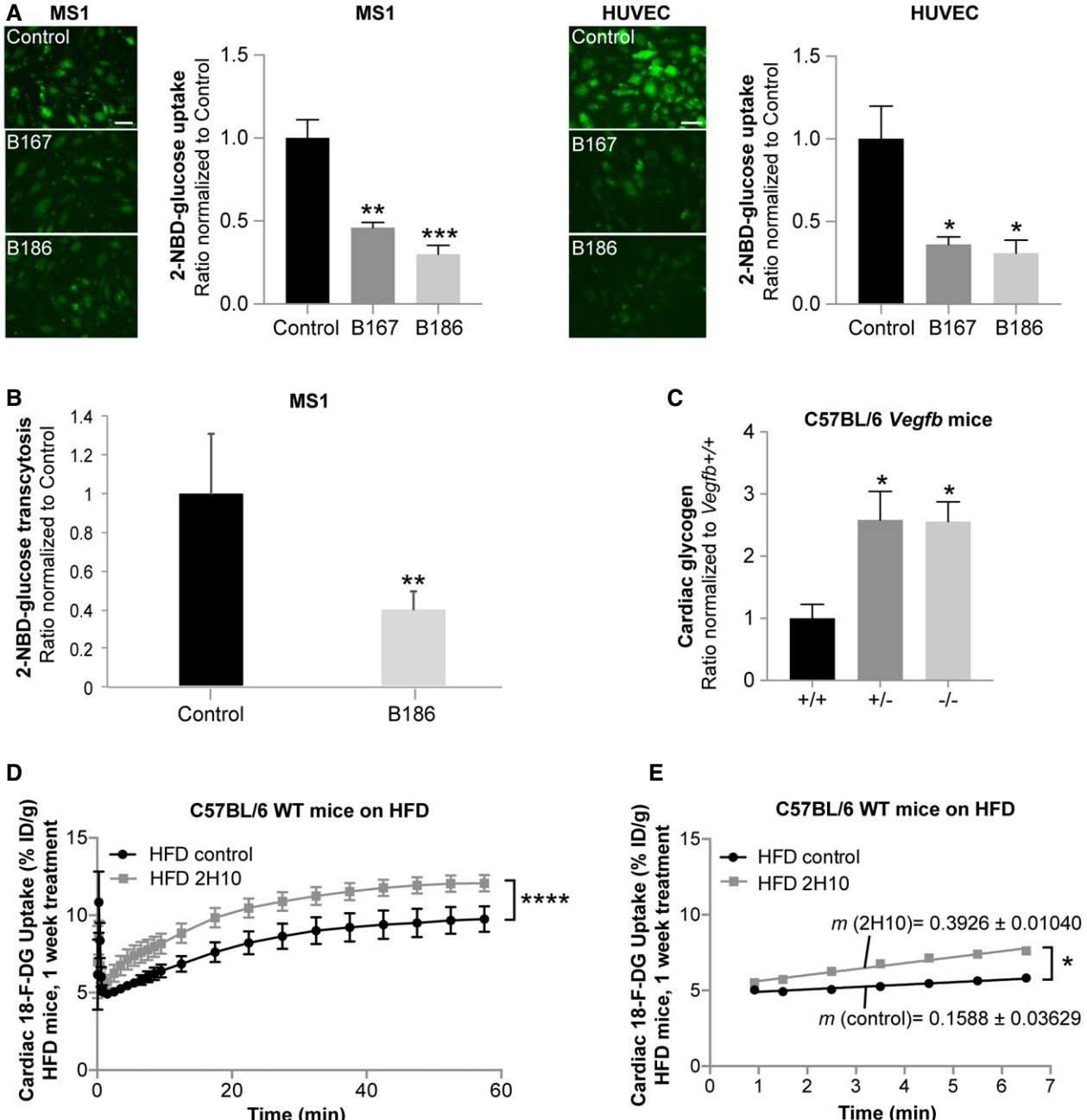

**Figure 1. VEGF-B regulates endothelial glucose uptake and transcytosis.**

A 2-NBD-glucose uptake in murine pancreatic endothelial cells (MS1) and primary human umbilical vein endothelial cells (HUVEC) treated for 2 h with VEGF-B$_{167}$ (B167) or VEGF-B$_{186}$ (B186) compared to control treated cells. Data presented as mean ± SEM from representative experiments performed in triplicates. Statistical evaluation using one-way ANOVA and Dunnett's multiple comparison test, P-value: *< 0.05, **< 0.01, ***< 0.001 (compared to untreated control). Scale bar, 20 μm.

B Trans-cellular 2-NBD-glucose transport in MS1 cells treated for 2 h with VEGF-B$_{186}$. Data presented as mean ± SEM from three individual experiments. Statistical evaluation using t-test, P-value: **< 0.01 (compared to untreated control).

C Cardiac glycogen content in 17-week-old male *Vegfb*$^{+/+}$ (n = 6), *Vegfb*$^{+/-}$ (n = 6), and *Vegfb*$^{-/-}$ mice (n = 5) presented as mean ± SEM. Statistical evaluation using one-way ANOVA and Dunnett's multiple comparison test, P-value: *< 0.05 (compared to *Vegfb*$^{+/+}$).

D Glucose uptake and accumulation ($^{18}$F-DG PET) in hearts of HFD-fed wild-type (WT) mice pre-treated for 1 week with 2H10 antibody (n = 4) compared to control antibody treated mice (n = 4). Data presented as mean ± SEM. Statistical evaluation using two-way ANOVA, P-value: ****< 0.0001.

E Linear regression analysis of the $^{18}$F-DG PET data in (D) assessing differences in initial (T = 0.9–6.5 min) glucose uptake rate. Data presented as mean curve slope (m) ± SEM. Statistical evaluation comparing slope differences using analysis of covariance (ANCOVA), P-value: *< 0.05.

Data information: See also Fig EV1.

insensitive *Slc2a1* glucose transporter (GLUT1) were observed (Fig 2A and Appendix Fig S1B). The identification of GLUT1 as the main EC glucose transporter was verified in human primary ECs (Fig 2B and Appendix Fig S1C). Stimulation of ECs with VEGF-B did not affect *SLC2A1* expression (Appendix Fig S2D), nor were *Slc2a* or *Slc5a* family members differentially expressed in cardiac ECs derived from *Vegfb*$^{-/-}$ mice (Fig 2A and Appendix Fig S1B).

VEGF-B signaling requires engagement with both VEGFR1 and NRP1 on ECs in order to increase FA uptake [20,21]. HUVECs express *FLT1* (VEGFR1), *NRP1* and considerable amounts of *SLC2A1*, but only minor amounts of endogenous *VEGF-B,* which makes them an appropriate model to study paracrine VEGF-B signaling and effects on glucose uptake (Fig 2C). siRNA-mediated knockdown of *FLT1*, *NRP1,* or *SLC2A1* all prevented VEGF-B-mediated reduction in EC glucose uptake (Fig 2D and Appendix Fig S1E), implying their involvement in regulation of endothelial glucose uptake. This was confirmed in MS1 ECs using stable shRNA-mediated knockdown of *Nrp1* or *Flt1* and inducible knockdown of *Slc2a1* (Fig EV2A and B). In contrast, knockdown of the receptor for VEGF-A, *KDR* (VEGFR2), had no impact on VEGF-B-dependent decrease in EC glucose uptake pinpointing the selectivity for VEGF-B downstream signaling (Fig 2D). Importantly, reduced endothelial glucose uptake capacity *per se* does not impair VEGF-B signaling. Phorbol 12-myristate 13-acetate treatment has previously been reported to block glucose uptake in tumor cells [33]. In HUVECs, phorbol 12-myristate 13-acetate correspondingly reduced glucose uptake to a similar degree as VEGF-B signaling alone, albeit the cell retained their responsiveness to a further decrease in glucose uptake following co-treatment with VEGF-B (Fig EV2C). Furthermore, GLUT1 reduction had no apparent effect on VEGF-B-mediated increase in FA uptake (Fig EV2D and E), whereas knockdown of *FLT1* or *NRP1* blunted the response (Fig EV2F).

Immunohistochemical analysis of murine cardiac tissue showed predominant GLUT1 staining in endothelial but also perivascular cells (Fig 2E). qPCR analysis on freshly isolated cardiac ECs compared to whole heart homogenates also confirmed expression of *Slc2a1* in ECs, together with *Flt1* and *Nrp1*, while *Vegfb* and *Slc2a4* (GLUT4) were predominantly expressed in non-endothelial cells (Fig 2F), collectively proposing an important role for endothelial GLUT1 in mediating glucose transport from the blood stream to muscular tissues *in vivo*.

To exclude the possibility that the observed decrease in glucose uptake in response to VEGF-B signaling is indirectly related to an altered cellular metabolism, the Seahorse® live-cell metabolic assay platform was utilized to assess energy substrate use and cellular energy balance. Intriguingly, HUVECs almost exclusively utilize glucose as energy substrate to produce ATP via glycolysis, both during starved and fed conditions (Fig EV3). Conversely, mitochondrial respiration and oxygen consumption were seemingly underutilized for ATP production purposes. Importantly, VEGF-B stimulation did not alter energy substrate utilization or total ATP production in the endothelial cells themselves, supporting a role of VEGF-B in energy substrate partitioning at the level of the vascular interphase and vascular transcytosis of energy substrates to the underlying tissue cells. Furthermore, unbiased transcriptional profiling of HBMECs stimulated with VEGF-B revealed no general effect on gene expression related to cell metabolism as summarized in Fig EV4.

Collectively, these data suggest specific signaling driven by VEGF-B rather than a compensatory response to changed substrate availability and energy status as conferred by the Randle cycle [15].

## VEGF-B signaling leads to changes in the plasma membrane microenvironment of GLUT1

Besides transcriptional regulation, GLUT1 activity and glucose uptake can be increased by recruitment of GLUT1 to the plasma membrane, similar to the regulation described for GLUT4 [34]. No obvious difference in the localization of GLUT1 and the early endosomal marker EEA1 was detected in VEGF-B stimulated HUVECs (Fig 3A). Also, total cellular GLUT1 protein content was unaltered in response to VEGF-B stimulation (Fig 3B). A more detailed analysis of the subcellular localization of GLUT1 and VEGF-B receptors in ECs was assessed by membrane separation and fractionation in a sucrose density gradient followed by Western blot analysis. GLUT1 primarily localized to the plasma membrane, VEGFR1 to the early endosome fraction, and NRP1 to the plasma membrane as well as several internal membranes, implicating a broader intracellular distribution pattern of NRP1 (Fig 3C and Appendix Fig S2A). VEGF-B stimulation did not evidently affect VEGFR1 distribution and only slightly shifted NRP1 from the plasma membrane to intracellular membranes. GLUT1 was still detected mainly in the

**Figure 2. VEGF-B regulated glucose uptake is mediated by GLUT1, NRP1, and VEGFR1.**

A  Relative mRNA expression of the *Slc2a* glucose transporter (GLUT) family members (*1-13*) in cardiac ECs derived from *Vegfb*$^{+/+}$ (*n* = 4) and *Vegfb*$^{-/-}$ (*n* = 4) mice. Data represent normalized PLIER values (mean ± SEM) from microarray analysis performed on *n* = 4+4 RNA samples (one RNA sample/mouse).

B  mRNA expression levels of the *SLC2A* transporter (GLUT) family members (*1-14*) in HUVECs. Data presented as mean ± StDev relative to *RPL19* expression from 3 independent experiments.

C  mRNA expression levels of *VEGFB, SLC2A1, NRP1,* and *FLT1* (VEGFR1) in HUVECs. Data presented as mean ± StDev relative to *RPL19* expression from 3 independent experiments.

D  2-NBD-glucose uptake in response to 2 h treatment with VEGF-B$_{186}$ (B186) in HUVECs exhibiting siRNA-mediated knockdown of *FLT1*, *NRP1*, *SCL2A1*, *KDR* (VEGFR2), or control siRNA. Data presented as mean ± SEM from three individual experiments. Statistical evaluation using one-way ANOVA and Sidak's multiple comparison test, *P*-value: *< 0.05 (comparisons made to respective untreated control).

E  Representative images of GLUT1 (left panels) and CD31 (right panels) immunohistochemistry performed on consecutive sections from paraffin embedded mouse hearts. Scale bar, 100 µm.

F  mRNA expression analysis from enriched cardiac endothelial cells (open columns, *n* = 4 mice) or whole heart (closed columns, *n* = 3 mice). Data presented as mean ± StDev relative to *Rpl19* expression. Enrichment factor is calculated as mean cardiac EC/mean whole heart and presented in the box.

Data information: See also Fig EV2 and Appendix Fig S1.

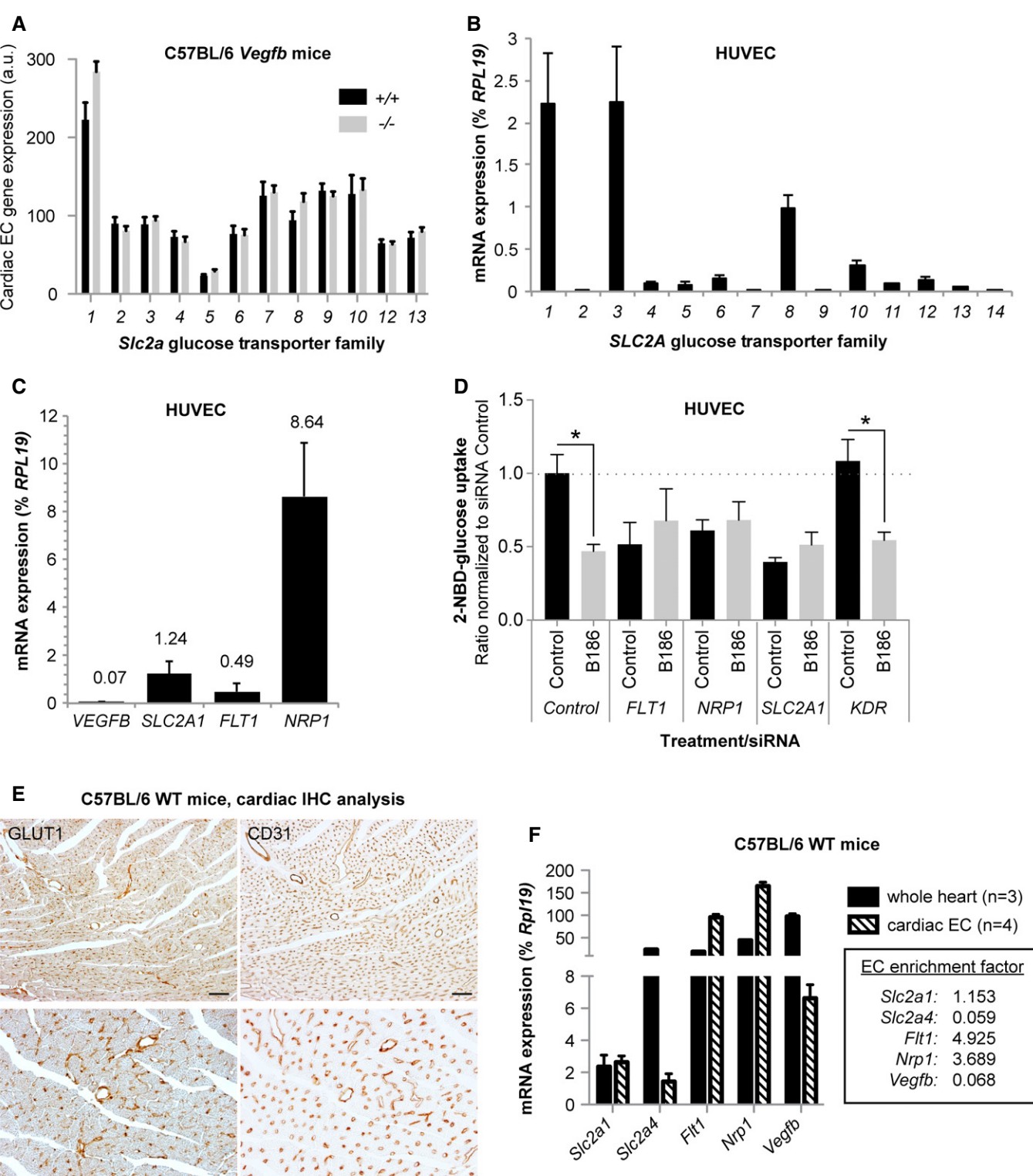

**Figure 2.**

plasma membrane fraction after VEGF-B stimulation, but was increased in fractions of higher density (Fig 3C). The main localization of GLUT1 in the plasma membrane and its increased association with higher density membrane fractions after VEGF-B

stimulation were confirmed with different plasma membrane marker proteins, with the best fit for the junction protein Connexin 43 (Appendix Fig S2B). A more detailed analysis of the plasma membrane fractions, representing 20–38% sucrose, revealed about

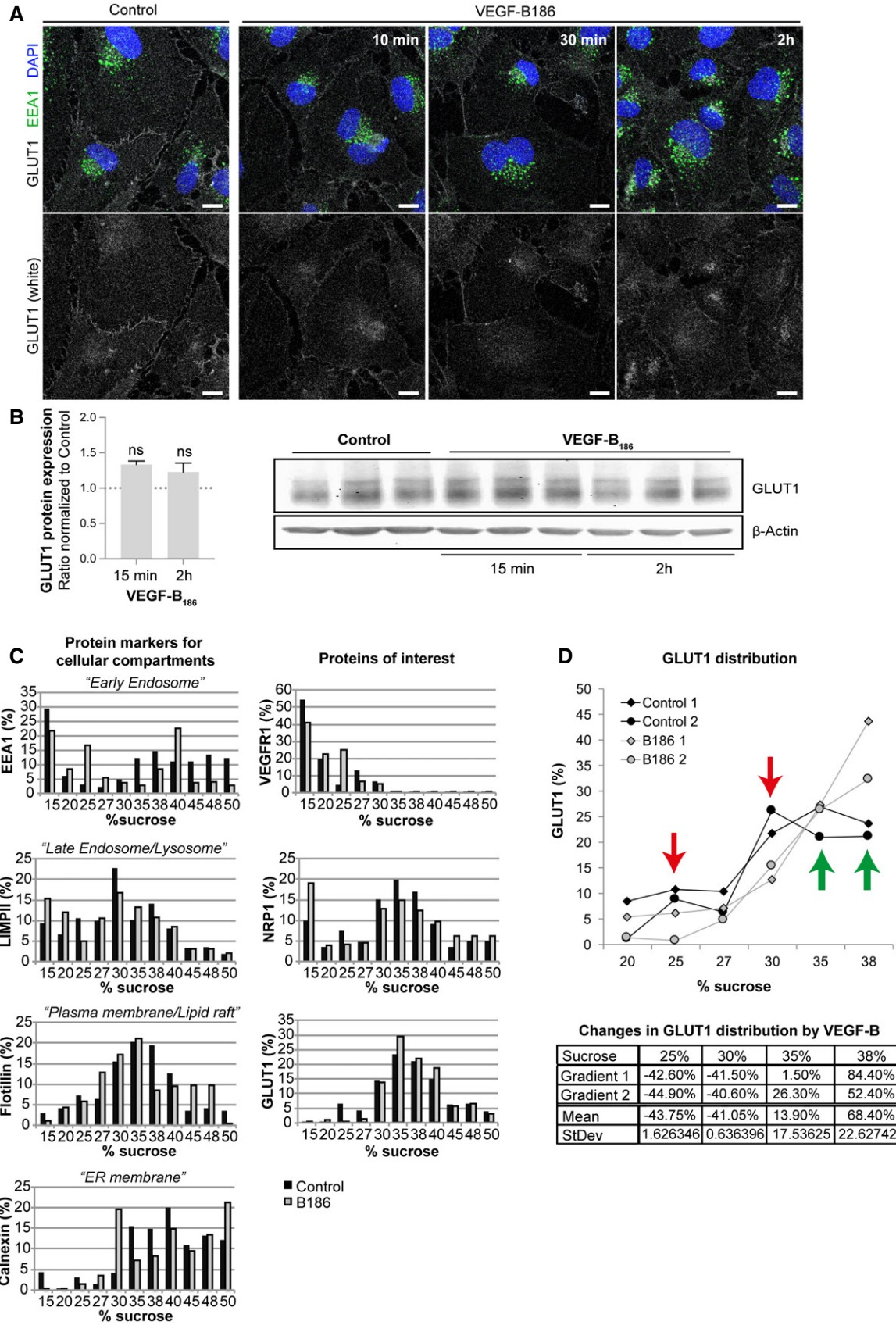

**Figure 3.**

◀

**Figure 3.  VEGF-B signaling changes microdomain localization of GLUT1 in the plasma membrane.**

A  Immunofluorescence labeling of GLUT1 (white), early endosomes (EEA1, green), and nuclei (blue) in HUVECs treated with VEGF-B$_{186}$ for 10, 30 min, or 2 h. Scale bar, 10 μm.

B  GLUT1 protein expression in HUVECs treated with VEGF-B$_{186}$ for 15 min or 2 h. Data presented as mean ± StDev of 3–4 independent experiments relative to β-Actin expression. Statistical evaluation using *t*-test revealed no significant changes. Western blots shown in right panel.

C  Sucrose gradient (15–50% sucrose) fractions of HUVEC cellular lysates after 2 h VEGF-B$_{186}$ stimulation were analyzed with Western blots shown in Appendix Fig S2. Different cellular sub-compartments were identified by accumulation of specific protein markers: EEA1 for early endosomes, LIMPII for late endosomes/lysosomes, flotillin for plasma membrane/lipid rafts, and calnexin for endoplasmic reticulum (ER) (left panels). Amount and subcellular distribution of VEGFR1, NRP1, and GLUT1 protein in control or VEGF-B$_{186}$ stimulated HUVECs shown in panels to the right. Data represent % protein relative to total protein of interest measured from a representative experiment.

D  Quantification of GLUT1 protein in % relative to total measured protein in HUVECs after 2 h VEGF-B$_{186}$ stimulation in different sucrose fractions (upper panel). Data from two independent experiments are shown. Green arrows mark fractions with increased GLUT1 content and red arrows mark fractions with decreased GLUT1 content after VEGF-B$_{186}$ stimulation. Changes in GLUT1 content in response to VEGF-B treatment are shown in the table (bottom panel).

Data information: See also Appendix Fig S2.

80% reduction of GLUT1 in less dense fractions (25 and 30% sucrose) and an equivalent increase in high-density fractions (representing 35 and 38% sucrose) in response to VEGF-B treatment (Fig 3D). A shift in protein distribution within the plasma membrane compartment has previously been observed in the context of lipid raft formation and recruitment of components critical for cell signal transduction [35–37]. Lipid rafts are highly ordered and densely packed cholesterol- and sphingolipid-rich membrane domains that exhibit a lower density [38]. The shift of GLUT1 to plasma membrane fractions with a higher density might thus reflect a change of the plasma membrane microenvironment and lipid rafts.

### VEGF-B signaling rapidly reduces endothelial cholesterol uptake

Differences in membrane density often reflect changes in the lipid composition. The shift of GLUT1 to higher density areas upon VEGF-B stimulation might thus reflect a change from a cholesterol/sphingolipid-rich to a cholesterol/sphingolipid-poor environment. In ECs, non-esterified cholesterol was primarily located to the plasma membrane, but also present in perinuclear vesicular structures, as judged by filipin staining (Fig 4A). Upon VEGF-B stimulation, the filipin staining for non-esterified cholesterol was decreased in ECs (Fig 4A), matching the observed shift in the density profile of the plasma membrane.

The temporal and sequential order of events regarding effects on FA/glucose uptake and non-esterified cholesterol content in response to VEGF-B signaling was resolved by time-chase studies in HUVECs. Cholesterol reduction was evident within 10 min after VEGF-B stimulation (Fig 4B), followed by a reduction in glucose uptake at 30 min (Fig 4C) and eventually increased FA uptake after 2 h of VEGF-B treatment (Fig 4D). The fast reduction of non-esterified cholesterol triggered by VEGF-B was further confirmed by quantitative lipid extraction (Fig 4E). Either decreased uptake or increased secretion of cholesterol can account for the observed decrease in non-esterified cholesterol. Decreased uptake of fluorescently labeled cholesterol (Topfluor-cholesterol) after VEGF-B stimulation however suggests a mechanism involving a reduction in uptake (Fig 4F). The reduced amount of non-esterified cholesterol in cellular membranes could also be a consequence of increased esterification and storage in lipid droplets. No differences in the levels of the major CE species were however detected following VEGF-B stimulation (Fig 4G).

### Depletion of endothelial plasma membrane cholesterol reduces glucose uptake

GLUT1 activity has previously been reported to be sensitive to the lipid environment in the plasma membrane [39–41]. The prompt reduction in endothelial cholesterol uptake and the change in the plasma membrane properties by VEGF-B may thus be functionally linked to the reduced glucose uptake. In line with this hypothesis, cholesterol extraction with methyl-beta-cyclodextrin (MbCD) to comparable levels as observed with VEGF-B stimulation was sufficient to evoke a similar reduction in EC glucose uptake as VEGF-B (Fig 5A). Inversely, reconstitution of membrane cholesterol to control levels blunted the effect of VEGF-B on glucose uptake (Fig 5B). Cholesterol overload (20 μg/ml) resulted in reduced basal glucose uptake (Fig 5B). This suggests the need for a tight control of plasma membrane cholesterol content to ensure optimal glucose uptake capacity. While cholesterol and glucose uptake seemed functionally linked in ECs, FA uptake was not affected by cholesterol depletion, suggesting that VEGF-B mediated glucose and FA uptake is mechanistically unrelated (Fig EV5A and B).

### VEGF-B reduces endothelial cholesterol content by interfering with LDLR recycling

The majority of circulating cholesterol is transported as CE inside lipoprotein particles, which are taken up by cells via receptor-mediated endocytosis. HUVECs express several lipoprotein-binding receptors, such as *SCARB1* (SRB1) and *LDLR* (Fig 6A). While the LDLR mediates uptake and transcytosis of LDL, SRB1 is involved in HDL transport [8,9]. siRNA-mediated knockdown of the *LDLR* blunted the effect of VEGF-B on endothelial cholesterol content as well as glucose uptake, whereas *SCARB1* knockdown had no effect, suggesting that the LDLR is involved in VEGF-B-mediated reduction of cholesterol and glucose uptake (Figs 6B and C, and EV5C). In contrast, FA uptake was not affected by *LDLR* knockdown (Fig EV5D). Freshly isolated lung ECs from *Ldlr*$^{-/-}$ mice displayed overall reduced cholesterol content and no further reduction after VEGF-B stimulation (Fig 6D), verifying an important function of the LDLR in endothelial cholesterol homeostasis. Together, these data suggest that VEGF-B-mediated increase in FA uptake is not reflecting a compensatory response to a reduced delivery of LDL-derived FAs, or altered membrane cholesterol composition, but rather represents an independent event. Reduced delivery of cholesterol from

exogenous sources could be compensated by increased *de novo* synthesis. Targeting the rate-limiting step in cholesterol biogenesis with simvastatin neither interfered with VEGF-B-mediated cholesterol and glucose uptake (Fig 6E), nor with the FA uptake response (Fig EV5E), suggesting that there is no compensation from internal sources within the studied time-frame.

To further explore how VEGF-B signaling influences LDLR function, the subcellular distribution of the LDLR was assessed. HUVECs stimulated with VEGF-B$_{167}$ or VEGF-B$_{186}$ for 20 min exhibited a

decrease in plasma membrane-bound LDLR, as determined by cell surface biotin labeling technique (Fig 6F). Immunofluorescence labeling confirmed a decreased cell surface LDLR expression (Fig 6G). This was accompanied by reduced binding, as well as uptake, of fluorescently labeled LDL (Dil-LDL) following VEGF-B stimulation, further pointing to an LDLR-dependent mechanism (Fig 6H and I, Appendix Fig S3).

No difference in mRNA expression or protein level of the LDLR was however detected following 15-min or 2-h VEGF-B stimulation,

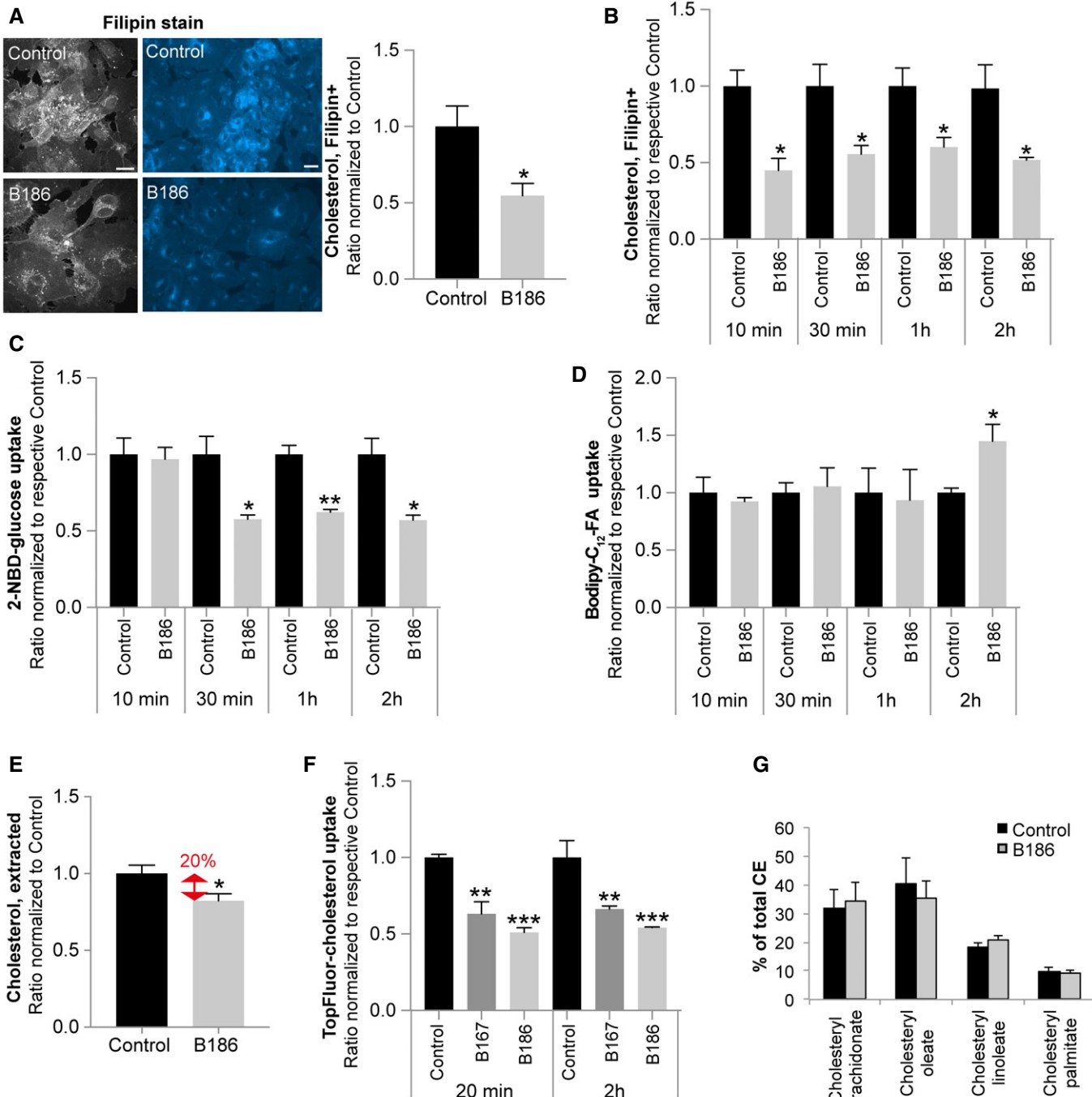

**Figure 4.**

◀

**Figure 4.  VEGF-B reduces membrane cholesterol prior to reduction of glucose uptake and increased FA uptake.**

A     Representative confocal (gray) and epifluorescence (blue) images of filipin staining in HUVECs in response to 2 h VEGF-B$_{186}$ stimulation (left panels). Cellular cholesterol content measured by quantification of fluorescent filipin staining. Data presented as mean ± SEM from a representative experiment performed in triplicates. Statistical evaluation using *t*-test, *P*-value: *< 0.05. Scale bars, 10 µm.

B–D   Measurement of cholesterol content (B), glucose uptake (C), and FA uptake (D) in HUVECs after VEGF-B$_{186}$ stimulation for different time points. Data presented as mean ± SEM from representative experiments performed in triplicates. Statistical evaluation using *t*-test, *P*-value: *< 0.05, **< 0.01, ***< 0.001 (compared to respective control).

E     Colorimetric quantification of non-esterified cholesterol extracted from HUVECs stimulated with VEGF-B$_{186}$ for 20 min. Data are presented as mean ± SEM of three individual experiments. Statistical evaluation using *t*-test, *P*-value: *< 0.05.

F     Uptake of a fluorescent cholesterol tracer (TopFluor) in HUVECs after VEGF-B$_{186}$ stimulation for 20 min or 2 h. Data presented as mean ± SEM from a representative experiment performed in triplicates. Statistical evaluation using one-way ANOVA and Dunnett's multiple comparison test, *P*-value: **< 0.01, ***< 0.001 (compared to respective untreated control).

G     Quantification of the major cholesteryl ester (CE) species in HUVECs after VEGF-B$_{186}$ stimulation for 20 min, analyzed by lipid mass spectrometry. Data presented as mean ± StDev of an experiment performed in triplicates.

Data information: See also Fig EV5.

rendering it unlikely that reduced translation or LDLR protein degradation accounted for the observed reduction in plasma membrane LDLR and membrane cholesterol content (Appendix Fig S4A and B). Evaluation of intracellular LDLR distribution using immunofluorescence staining demonstrated enrichment in perinuclear clusters (Appendix Fig S4C) in areas adjacent to the Golgi compartment cisternae labeled with GM130 (Appendix Fig S4E), and partially co-localized with the early endosome marker EEA1 and lysosomal marker LAMP1 (Appendix Fig S5) under basal growth conditions. This is in line with previous studies in liver, where LDL/LDLR complexes are endocytosed and transported to the Golgi and lysosomal compartments [6], after recycling of the receptor back to the plasma membrane. VEGF-B stimulation correlated with reduced clustering around the Golgi compartment and instead the LDLR displayed a dispersed intracellular distribution pattern (Appendix Figs S4C–E and S5). Collectively, these data imply that VEGF-B signaling interferes with LDLR recycling to the plasma membrane leading to reduced plasma membrane LDLR localization, correlating with reduced LDL-cholesterol binding and subsequent uptake, as a mechanism of reduced endothelial cholesterol content.

### VEGF-B regulates cardiac cholesterol content *in vivo*

The effect of VEGF-B on membrane cholesterol content could be confirmed *in vivo* in different mouse models. *Vegfb*$^{+/-}$ and *Vegfb*$^{-/-}$ mice exhibited increased vessel-associated cholesterol in the heart compared to littermate wild-type control mice, as determined by fil-ipin co-staining with the endothelial apical marker protein podoca-lyxin (Fig 7A and Appendix Fig S6A). This increase in vessel-associated cholesterol was accompanied by an overall increased level of non-esterified cholesterol throughout the heart (Fig 7B), possibly linked to enhanced endothelial cholesterol uptake from blood. Hearts from *Flt1*$^{+/-}$ mice [21] also displayed increased membrane cholesterol content compared to wild-type mice (Fig 7C), supporting a role of VEGF-B signaling in endothelial cholesterol transport and cardiac cholesterol supply *in vivo*.

### VEGF-B inhibition restores cardiac cholesterol homeostasis in diabetes models

Inhibition of VEGF-B has been shown to be beneficial in genetic and HFD-induced models of diabetes [17]. Leptin receptor-deficient mice (*Lepr*$^{db/db}$, hereafter denoted *db/db*) develop overt obesity,

dyslipidemia, insulin resistance, and T2DM [42,43]. Homozygous *db/db* diabetic mice displayed decreased cardiac content of non-esterified cholesterol compared to *db/+* and wild-type mice despite development of obesity (Fig 7D). This is in line with observations in rats with type 1 diabetes mellitus, reported to exhibit a decreased level of non-esterified cholesterol compared to CE in the heart, as well as other organs [44]. The HFD model of T2DM [45] correspondingly exhibited decreased cardiac levels of non-esterified cholesterol, compared to age-matched littermates on standard diet (Fig 7E). *Vegfb*$^{+/-}$ or *Vegfb*$^{-/-}$ *db/db* mice displayed a gene-dose-dependent increase in non-esterified cholesterol content in the heart, compared to *Vegfb*$^{+/+}$ *db/db* mice (Fig 7D). In accordance, long-term (20 week) treatment with anti-VEGF-B antibodies (2H10) in HFD mice resulted in restored cardiac levels of non-esterified cholesterol, comparable to levels obtained in lean control mice (Fig 7E). To determine whether VEGF-B directly impact cardiac cholesterol homeostasis in diabetic settings, VEGF-B signaling was acutely blocked with anti-VEGF-B antibodies (2H10) in HFD-fed mice. In response to a short treatment (1 week) with 2H10, membrane cholesterol levels increased, suggesting that the changes in tissue cholesterol are directly consequential to VEGF-B signaling (Fig 7F).

As no differences in expression of genes involved in cholesterol homeostasis were observed in *Vegfb*$^{-/-}$ compared to *Vegfb*$^{+/+}$ hearts (Appendix Fig S6B), it can be assumed that EC cholesterol uptake and differences in transendothelial cholesterol transport may be an underlying mechanism for the observed changes in tissue cholesterol levels. In summary, our data propose a novel function of VEGF-B in controlling tissue cholesterol homeostasis by regulating endothelial LDL-cholesterol uptake and transport. The beneficial effect that VEGF-B targeting has on diabetes might thus in part involve restoration of membrane cholesterol content with impact on multiple signaling and transport systems involved in diabetes.

## Discussion

This study identified a relationship between plasma membrane cholesterol content and glucose transport capacity in ECs that can be regulated by VEGF-B signaling. Diminished glucose uptake as a consequence of plasma membrane cholesterol depletion has previously been observed in cancer cell lines, adipocytes and skeletal muscle cells [39–41]. In the aforementioned studies, the reduction in membrane cholesterol was achieved by chemical extraction with

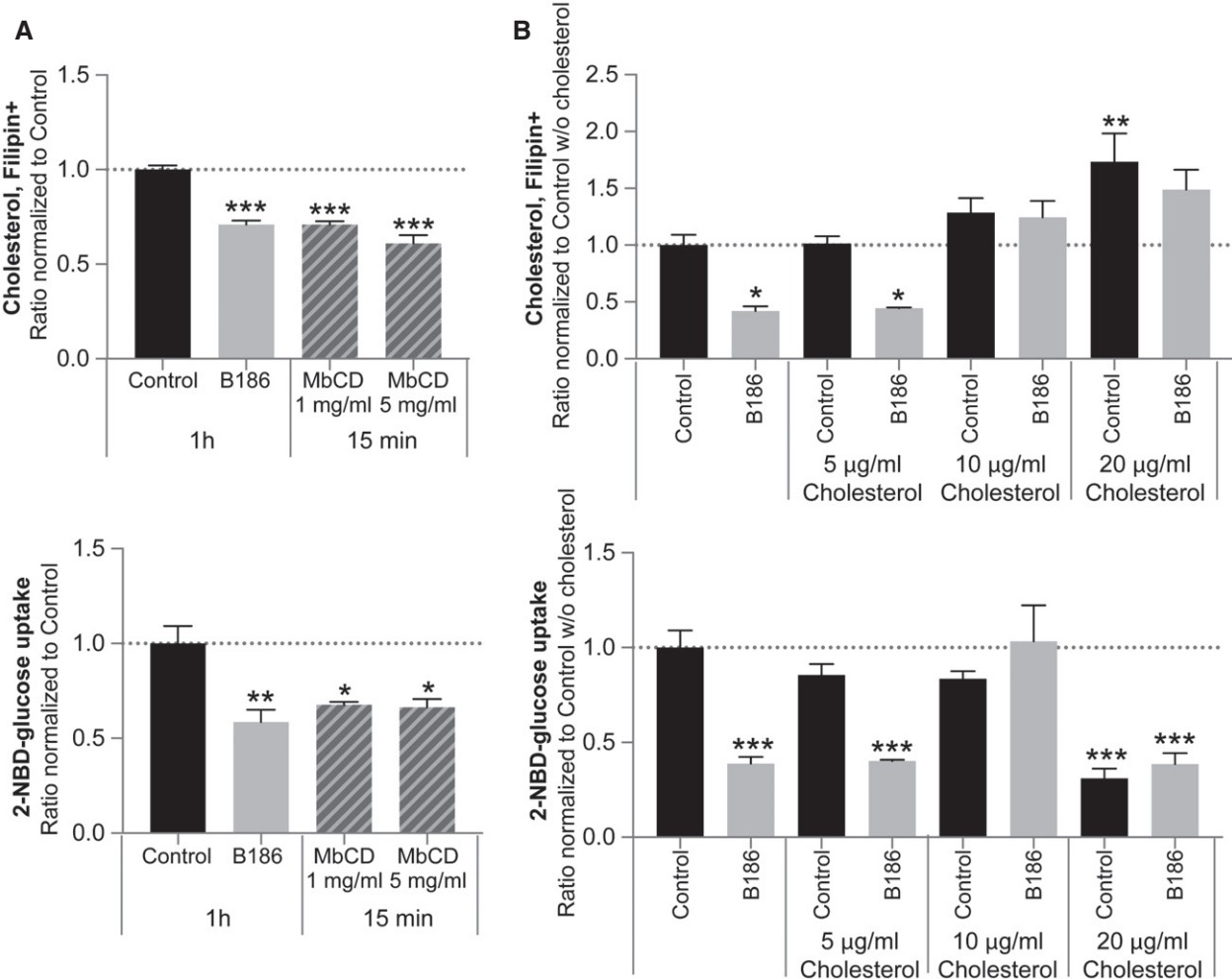

**Figure 5. Cholesterol content and glucose uptake are tightly linked in ECs.**

A  Measurement of cellular cholesterol content (upper panel) and glucose uptake (lower panel) in HUVECs treated with VEGF-B$_{186}$ for 1 h or after 15-min treatment with cholesterol-extracting methyl-beta-cyclodextrin (MbCD). Data are presented as mean ± SEM from a representative experiment performed in triplicates. Statistical evaluation using one-way ANOVA and Dunnett's multiple comparison test, $P$-value: ***< 0.001 (compared to untreated control).

B  Measurement of cellular cholesterol content (upper panel) and glucose uptake (lower panel) in HUVECs treated with VEGF-B$_{186}$ for 15 min alone or supplemented with soluble methyl-beta-cyclodextrin complexed with cholesterol (5, 10 or 20 µg/ml cholesterol). Data presented as mean ± SEM from a representative experiment performed in triplicates. Statistical evaluation using one-way ANOVA and Dunnett's multiple comparison test, $P$-value: *< 0.05, **< 0.01, ***< 0.001 (compared to control w/o cholesterol).

Data information: See also Fig EV5.

methyl-beta-cyclodextrin or arose as a result of statin treatment and impaired *de novo* cholesterol synthesis, [40,41]. In contrast, we here revealed a mechanism independent of *de novo* synthesis, by which VEGF-B signaling in ECs reduces cell surface LDLR expression and alters intracellular distribution of the LDLR, possibly related to impaired recycling of the LDLR back to the plasma membrane; correlating with impeded uptake of exogenous cholesterol. In line with previous findings in other cell types, changes in membrane cholesterol content impacted GLUT1-facilitated glucose transport also in ECs. In addition, our data verified GLUT1 as the main endothelial glucose transporter, ubiquitously expressed in ECs from various types of vascular beds, including the blood-brain barrier [1]. The data furthermore implicates an insulin-independent glucose uptake and transport mechanism in ECs [32].

Additionally, we demonstrated the relevance of a tightly controlled cholesterol homeostasis in ECs. Suggestively, optimal GLUT1 activity is determined by the lipid composition of the plasma membrane, a feature which might be shared by all GLUT1 expressing cell types. Recently, a cholesterol-binding motif in GLUT1 was identified [40,41]. The cholesterol recognition/interaction amino acid consensus (CRAC) sequence locates to transmembrane domain 2 and 8 of GLUT1 in the vicinity of the genetic mutations G91D and R33W, which affect GLUT1 membrane topology [46]. Crystallization of GLUT1 suggests a rocker-switch-like mechanism for glucose transport, comprising a cycle of outward open, outward ligand-occluded, inward open, and inward ligand-bound conformation [47]. Mutation in E329Q within the CRAC motif stabilizes the inward facing conformation [47,48]. Thus, interaction with

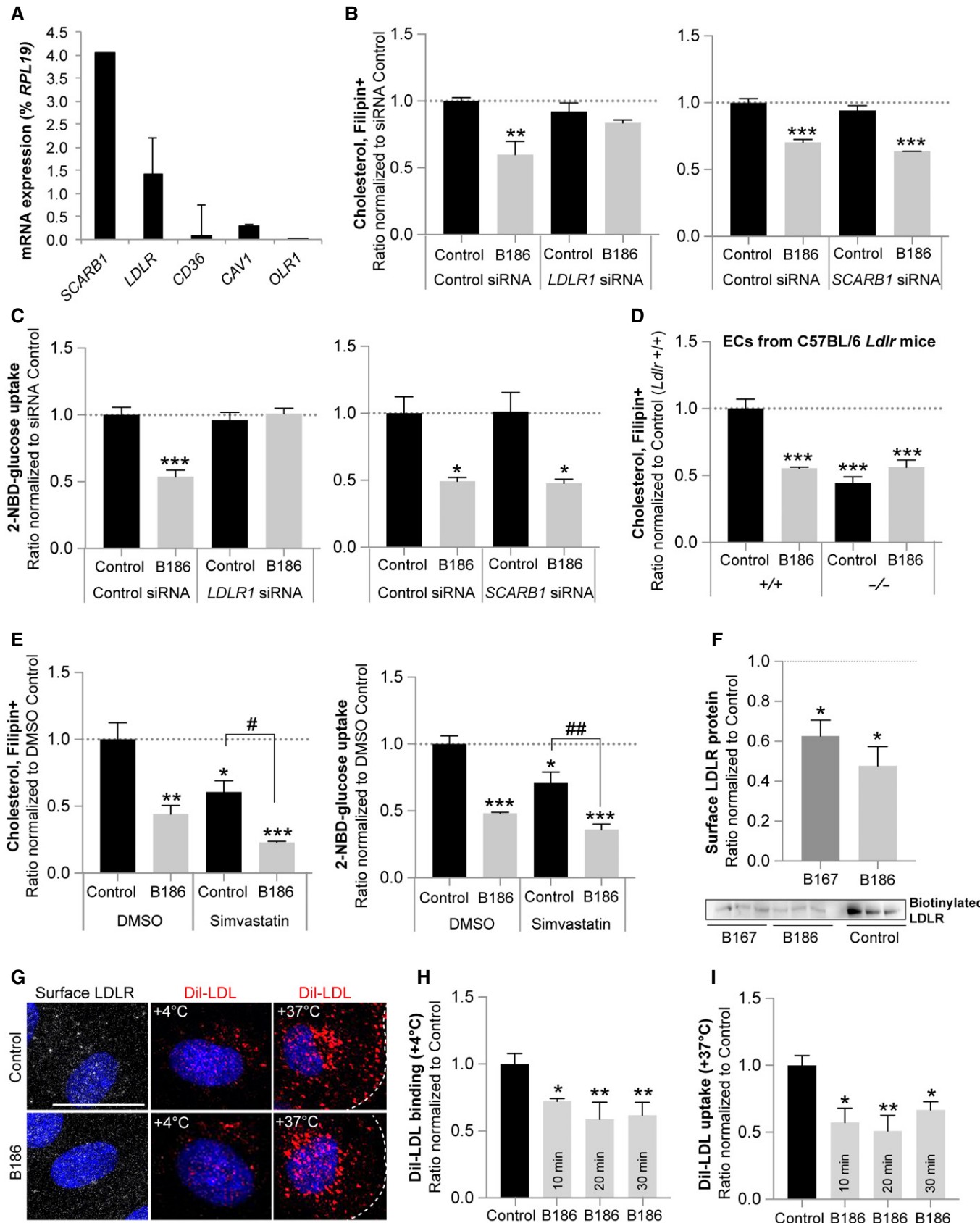

Figure 6.

**Figure 6.  VEGF-B regulates endothelial cholesterol handling in an LDL receptor-dependent manner.**

A   mRNA expression analysis of lipoprotein receptors in HUVECs. Data presented as mean ± StDev relative to *RPL19* expression from 3 independent experiments.

B, C   Cellular cholesterol content (B) and glucose uptake (C) in HUVECs treated with control siRNA or siRNA targeting *LDLR* or *Scavenger receptor B1* (*SCARB1*, SRBI) followed by 15 min stimulation with VEGF-B$_{186}$. Data presented as mean ± SEM from representative experiments performed in triplicates. Statistical evaluation using one-way ANOVA and Dunnett's multiple comparison test, *P*-value: *< 0.05, **< 0.01, ***< 0.001 (compared to respective control).

D   Cellular cholesterol content of ECs freshly isolated from *Ldlr*$^{+/+}$ and *Ldlr*$^{-/-}$ mice stimulated for 15 min with VEGF-B$_{186}$. Data presented as mean ± SEM from a representative experiment performed in triplicates. Statistical evaluation using one-way ANOVA and Tukey's multiple comparison test, *P*-value: ***< 0.001 (compared to *Ldlr*$^{+/+}$ control).

E   Cellular cholesterol content (left panel) and glucose uptake (right panel) of HUVECs treated with *de novo* cholesterol synthesis inhibitor simvastatin. Data presented as mean ± SEM from a representative experiment performed in triplicates. Statistical evaluation using one-way ANOVA and Tukey's multiple comparison test, *P*-value: *< 0.05, **< 0.01, ***< 0.001 (compared to untreated control). *P*-value: #< 0.05, ###< 0.001 (compared to simvastatin control).

F   Cell surface biotinylated LDLR protein levels in HUVECs after 20 min of VEGF-B$_{167}$ or VEGF-B$_{186}$ stimulation. Data presented as mean ± SEM from three independent experiments. Western blot shown below. Statistical evaluation using *t*-test, *P*-value: *< 0.05 (compared to control).

G   Representative images of cell surface distribution of the LDLR by immunofluorescence staining of non-permeabilized cells (white, left panels), Dil-LDL cell surface binding (evaluated at 4°C, middle panels), and Dil-LDL uptake (evaluated at 37°C, right panels) in HUVECs in response to VEGF-B$_{186}$ stimulation for 10 min. Scale bar, 10 μm. Overview pictures are found in Appendix Fig S3.

H, I   Quantification of Dil-LDL binding (H) and uptake (I) in HUVECs stimulated with VEGF-B$_{186}$ for 10 min, 20 min and 30 min. Data presented as mean ± SEM from a representative experiment performed in triplicates. Statistical evaluation using one-way ANOVA and Dunnett's multiple comparisons test, *P*-value: *<0.05, **<0.01 (compared to untreated control).

Data information: See also Appendix Figs S3–S5.

membrane cholesterol might be important to overcome the energy barrier for transition to the outward facing conformation. Along these lines, super-resolution spatial imaging has provided evidence that clustering of GLUT1 in lipid raft microdomains is disrupted after methyl-beta-cyclodextrin treatment [37]. Future studies are therefore encouraged to resolve the particularities of spatial GLUT1 microdomain expression and consequence for glucose uptake.

In particular, we show that the pool of non-esterified cholesterol in the plasma membrane can be promptly reorganized in response to VEGF-B signaling. Hence, our data might suggest a mechanistic relationship between EC- and parenchymal cholesterol content, and impact on glucose availability within a tissue. We specifically demonstrated that VEGF-B signaling obstructed LDL-cholesterol binding and uptake into ECs through decreased plasma membrane LDLR localization, suggesting a cell intrinsic interaction between VEGF-B and the LDLR pathway. Indeed,

VEGFR1 has previously been shown to co-endocytose together with the LDLR upon LDL binding, directly influencing VEGFR1 downstream signaling [49].

The observed link between endothelial uptake and transcytosis of circulating cholesterol and glucose may represent a common feature underlying development and progression of diabetes and diabetic complications. The findings give additional support to the well-established correlation between dyslipidemia and altered lipoprotein metabolism, and the development of insulin resistance and reduced tissue glucose uptake [50]. As proposed, endothelial insulin resistance and reduced interstitial insulin levels precede insulin resistance in skeletal muscle, liver, and adipose tissues [51,52]. Thus, when insulin-driven glucose uptake and interstitial insulin are reduced, the more crucial it will be to ensure that insulin-independent transendothelial glucose supply via GLUT1 does not become a limiting factor.

**Figure 7.  VEGF-B regulates cardiac cholesterol content *in vivo*.**

A   Quantification of non-esterified cholesterol content by means of filipin staining in the blood vessel compartment (podocalyxin+) in hearts of 15-week-old male C57BL/6 *Vegfb*$^{+/-}$ (*n* = 5) and *Vegfb*$^{-/-}$ mice (*n* = 7), normalized to wild-type (*Vegfb*$^{+/+}$) mice (*n* = 8). Data presented as box-and-whisker plots. Boxes represent lower/upper quartiles with the median values indicated with a horizontal line. Whiskers represent min–max values. Statistical evaluation using one-way ANOVA and Dunnett's multiple comparison test, *P*-value: *< 0.05, **< 0.01 (compared to *Vegfb*$^{+/+}$ mice).

B   Quantification of non-esterified cholesterol content by means of filipin staining in hearts of 15-week-old male C57BL/6 *Vegfb*$^{+/-}$ (*n* = 5) and *Vegfb*$^{-/-}$ (*n* = 7) mice, normalized to wild-type (*Vegfb*$^{+/+}$) mice (*n* = 8). Data presented as box-and-whisker plots. Boxes represent lower/upper quartiles with the median values indicated with a horizontal line. Whiskers represent min–max values. Statistical evaluation using one-way ANOVA and Dunnett's multiple comparison test, *P*-value: **< 0.01 (compared to *Vegfb*$^{+/+}$ mice).

C   Quantification of non-esterified cholesterol content by means of filipin staining in hearts of 25-week-old male CD1 *Flt1*$^{+/lacz}$ (*n* = 20) mice, normalized to wild-type CD1 (*Flt1*$^{+/+}$) mice (*n* = 16). Data presented as box-and-whisker plots. Boxes represent lower/upper quartiles with the median values indicated with a horizontal line. Whiskers represent min–max values. Statistical evaluation using *t*-test, *P*-value: **< 0.01.

D   Quantification of non-esterified cholesterol content by means of filipin staining in hearts of 12- to 16-week-old male C57BL/6 wild-type (*Vegfb*$^{+/+}$) (*n* = 4), BKS.C57BL/6 *db/+ Vegfb*$^{+/+}$ (*n* = 8), *db/db Vegfb*$^{+/-}$ (*n* = 10), and *db/db Vegfb*$^{-/-}$ (*n* = 4) mice, normalized to *db/db Vegfb*$^{+/+}$ (*n* = 6) mice. Data presented as box-and-whisker plots. Boxes represent lower/upper quartiles with the median values indicated with a horizontal line. Whiskers represent min–max values. Statistical evaluation using one-way ANOVA and Dunnett's multiple comparison test, *P*-value: *< 0.05, ***< 0.001 (compared to *db/db Vegfb*$^{+/+}$ mice).

E   Left panel: Quantification of non-esterified cholesterol content by means of filipin staining in hearts of C57BL/6 male mice after 25 weeks on HFD, long-term treated with isotype control (*n* = 17) or VEGF-B blocking antibody (2H10, *n* = 17) initiated after 5 weeks on HFD, normalized to age-matched mice on chow diet (*n* = 10). Data presented as box-and-whisker plots. Boxes represent lower/upper quartiles with the median values indicated with a horizontal line. Whiskers represent min–max values. Right panel: Body weights (mean ± SEM) of HFD (*n* = 17+17) and chow fed littermate mice (*n* = 10). Statistical evaluation using one-way ANOVA and Dunnett's multiple comparison test, *P*-value: *< 0.05, ***< 0.001 (compared to HFD control mice).

F   Quantification of non-esterified cholesterol content by means of filipin staining in hearts of mice on HFD for 17 weeks followed by treatment for 1 week with 2H10 antibody (*n* = 5) compared to control antibody treated mice (*n* = 5). Data presented as box-and-whisker plots. Boxes represent lower/upper quartiles with the median values indicated with a horizontal line. Whiskers represent min–max values. Statistical evaluation using *t*-test, *P*-value: **< 0.01.

Data information: See also Appendix Fig S6.

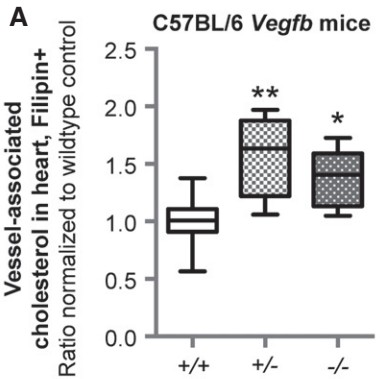

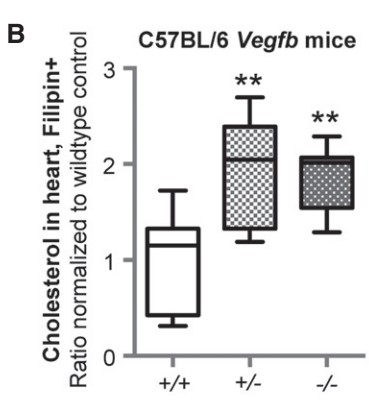

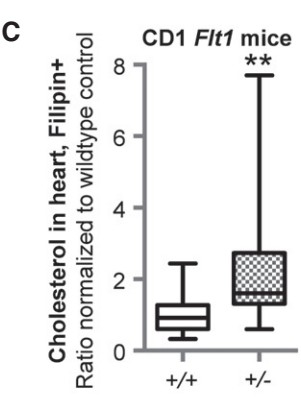

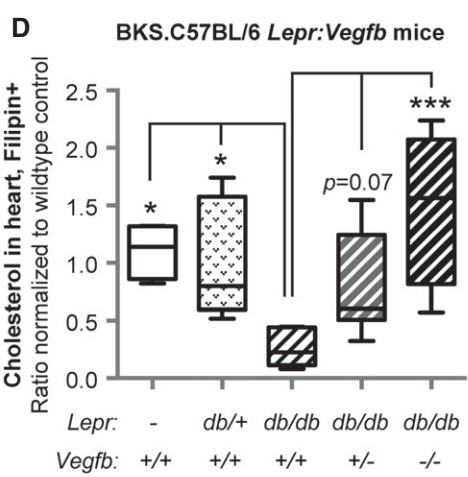

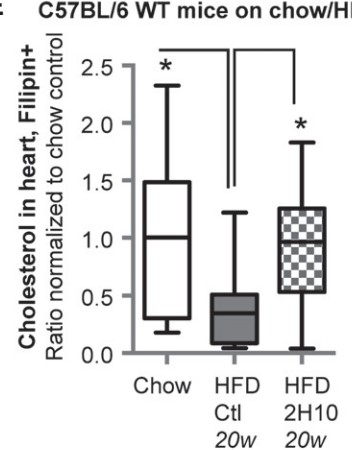

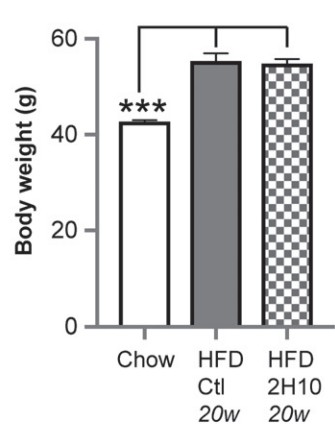

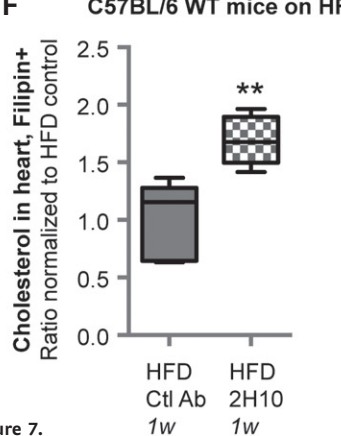

**Figure 7.**

We have previously reported that VEGF-B increases FA uptake and transcytosis in ECs [20]. According to the Randle cycle theorem, cells reduce glucose usage when lipids are available and *vice versa* [15]. Notably, stimulation with VEGF-B reduced LDLR-mediated cholesterol uptake within minutes, immediately followed by a reduction in glucose uptake, while the increase in FA uptake was evident at the earliest a few hours after VEGF-B administration. We could furthermore show that FA uptake was not dependent on plasma membrane cholesterol content. Instead, VEGF-B signaling in ECs seemed to independently control EC transcytosis of glucose and FAs, respectively. We could furthermore exclude an indirect effect on glucose or fatty acid uptake related to altered substrate utilization and energy homeostasis within the ECs.

Another interesting aspect involved the shift of lipid source in response to VEGF-B signaling. While LDL particles are supplied from the liver, FAs are derived mainly from lipolysis of stored TAG in adipose tissue. Especially, non-esterified FAs (NEFAs) have been suggested to be involved in diabetes and diabetic complications

[15,53–58]. Increased plasma NEFAs contribute to ectopic lipid accumulation in tissue, a shared hallmark of, *e.g.,* muscle insulin resistance [59], chronic kidney disease [60], and diabetic nephropathy [19]. This strengthens the notion that limiting VEGF-B signaling could constitute a potential treatment in metabolic diseases, where such a switch in lipid sources is detrimental [17–19].

Even though acute VEGF-B signaling evokes abrupt changes in cholesterol uptake, the relatively stable and constitutive tissue expression of VEGF-B suggests a homeostatic function. Hence, a proposed physiological role of VEGF-B is to limit excessive LDL-cholesterol uptake and plasma membrane cholesterol loading. Slight skewing of membrane cholesterol homeostasis could conversely infer defective receptor signaling and dysregulate nutrient transport, setting the path for metabolic disease progression. Interestingly, transcription of the *Vegfb* gene is upregulated in diabetic muscle tissues [17,18]. This notion is consistent with our findings in obese and/or T2DM mice, as well as previous observations in rats and rabbits with type 1 diabetes mellitus [44,61], verifying lower tissue levels of non-esterified cholesterol in diabetic mice compared to healthy controls. Membrane cholesterol content was effectively reconstituted when inhibiting VEGF-B signaling, indicating increased LDL-cholesterol uptake from blood and transcytosis to the underlying tissue. Of note, the accumulation of neutral lipids, *e.g.,* TAG and presumably also CE, in tissues of obesity-driven T2DM suggests sufficient availability of cholesterol. Instead of maintaining cholesterol homeostasis in the plasma membrane, cholesterol may then preferably esterify with incoming NEFAs. CE levels in ECs were however not altered by VEGF-B stimulation, supporting a role for VEGF-B in endothelial LDL uptake and cholesterol transcytosis. Cardiac *de novo* cholesterol synthesis furthermore appeared unaffected in *Vegfb*-deficient mice, pointing to a mechanism at the level of LDL-cholesterol uptake from blood, seemingly independent on FA uptake from TAG-rich lipoproteins or circulating NEFAs.

Changes in membrane cholesterol impact diverse processes. It is known that the function of different receptors, transmembrane or peripheral membrane proteins, is modulated by the lipid environment and their association to lipid rafts. This involves signaling in ECs themselves, *e.g.,* adherence junctions [62], VEGFR2 signaling [63] and angiogenesis [64], or within the parenchyma, *e.g.,* insulin signaling in neurons [65], as well as in pathological conditions like cancer [66,67]. VEGF-B has been reported to affect different processes like neuronal survival [68,69], arterial growth [23–25], or influencing VEGF-A signaling [23], effects which in part could be related to its effect on membrane cholesterol homeostasis.

In conclusion, our study provides mechanistic insight into VEGF-B signaling in ECs and postulates new clues on how regulation of EC metabolism and nutrient transport by VEGF-B are linked to different physiological and pathological processes. Our results point to a role of the vasculature, and specifically EC cholesterol homeostasis, as a novel concept to understand and treat diabetes and its co-morbidities.

# Materials and Methods

For details regarding materials, antibodies, primers, and probes; see Appendix.

## Cell culture

Cells were grown on 1% gelatin-coated tissue culture (TC)-treated dishes (BD Falcon) at 37°C and 5% $CO_2$ in Endothelial Cell Growth Medium 2 (PromoCell) (for HUVEC and HBMEC) or Endothelial Cell Growth Medium MV2 (PromoCell) (for MS1 and HCMEC) supplemented with the respective SupplementMix (containing growth factors and 2 or 5% FBS, respectively) and 10 Units/ml penicillin/streptomycin.

## VEGF-B stimulation

Cells were starved for 2 h prior to VEGF-B stimulation in Endothelial Cell Basal Medium (PromoCell) supplemented with 2.5% FBS (for cholesterol and glucose uptake assays) or 0.1% fatty acid-free bovine serum albumin (0.1% BSA) (for FA uptake assay). Then, 100 ng/ml mouse VEGF-B$_{167}$ or VEGF-B$_{186}$ in PBS-0.1% BSA or the same volume of plain PBS-0.1% BSA (Control) was added for the indicated time period in starvation medium (Endothelial Cell Basal Medium supplemented with 2.5% FBS or 0.1% BSA). In case of inhibitor studies, cells were starved for 1.5 h and then incubated for another 30 min with inhibitor in starvation medium. The VEGF-B stimulation was thereafter performed in the presence of the inhibitor.

## Glucose uptake assay

Cells were incubated in starvation medium [Endothelial Cell Basal Medium 2/MV2 supplemented with 2.5% FBS] for 2 h before VEGF-A, VEGF-B (both 100 ng/ml), or vehicle (Control) stimulation for indicated time points. Thereafter, cells were washed with Krebs–Ringer solution without glucose [120 mM NaCl, 24 mM $NaHCO_3$, 4.8 mM KCl, 1.2 mM $MgCl_2$, 1.2 mM $KH_2PO_4$, and 5 mM HEPES, pH 7.4] for 10 min at 37°C, 5% $CO_2$. Next, cells were incubated for 20 min at 37°C, 5% $CO_2$ in starvation medium supplemented with 5 mM L-glucose and 1 mM 2-NBD-glucose. Cells were subsequently washed 3× with PBS supplemented with 5 mM D-glucose and fixed with 4% paraformaldehyde in PBS (4% PFA) supplemented with 5 mM D-glucose and immediately subjected to evaluation and image capture using a Zeiss Axio Observer Z1 inverted microscope equipped with a 20× long distance LD Plan+-Neofluar fluorescence objective (N/A = 0.4 Ph2 Korr $\propto$/1.5), an AxioCam MRm fluorescence CCD camera and ZEN 2009 software (Carl Zeiss Microimaging GmbH). Each condition was performed in triplicates, and 15 images in total were taken per condition. Images were analyzed with ImageJ/Fiji software by setting threshold and measuring the integrated density per frame. Mean values were normalized to Control (set to 1).

## Fatty acid uptake assay

Cells were incubated in starvation medium [Endothelial Cell Basal Medium 2/MV2 supplemented with 0.1% fatty acid-free BSA (0.1% BSA)] for 2 h before VEGF-B (100 ng/ml) or vehicle (Control) stimulation for indicated time points. Thereafter, cells were washed with PBS-0.1% BSA and incubated with 20 μM Bodipy-$C_{12}$-FA in PBS-0.1% BSA for 5–10 min at 37°C, 5% $CO_2$. In Fig 4D, cells were starved in Endothelial Cell Basal Medium 2 supplemented with

2.5% FBS instead of 0.1% BSA. Cells were subsequently washed 3× with PBS-0.1% BSA and fixed with 4% PFA. Cells were immediately evaluated and imaged as described above for the glucose uptake assay.

**Transwell assay**

Permeable transwell inserts (BD Falcon) with 0.4 μm pore size suitable for 24 well plates were coated with Matrigel. MS1 cells were seeded onto the Matrigel matrix ($1 \times 10^5$ cells in 100 μl). After adherence, growth medium (Endothelial Cell Growth Medium MV2) was added to the upper and lower chamber and cells were grown o/n. The next day, the procedure was repeated (another $1 \times 10^5$ MS1 cells in 100 μl were seeded onto the insert). Before the experiment, adequate tightness of the cell monolayer was confirmed by measuring the TEER (Transendothelial Electrical Resistance) with a Milli-cell-ERS electrode from Millipore (Billerica, USA). Only wells with at least 60 $\Omega/cm^2$ were selected, and three transwell inserts with the same mean resistance per condition were used. Cells were incubated in starvation medium [Endothelial Cell Basal Medium MV2 supplemented with 2.5% FBS] for 2 h before stimulation with 100 ng/ml VEGF-B$_{186}$ or vehicle (Control) for 2 h (added to the lower chamber). Thereafter, cells were washed for 10 min with Krebs–Ringer solution without glucose. Next, the medium in the lower chamber was exchanged with Krebs–Ringer solution without glucose containing vehicle (Control) or VEGF-B$_{186}$ (100 ng/ml), while the upper chamber was filled with starvation medium supplemented with 5 mM L-glucose and 1 mM 2-NBD-glucose. Aliquots of 10 μl were taken from the lower chamber 20 min after adding the 2-NBD-glucose tracer to the upper chamber, mixed with 90 μl $H_2O$ in a 96 well plate and analyzed for fluorescence in a plate reader (POLAR-star Omega from BMG Labtech).

**Low-density lipoprotein uptake assay**

HUVECs were grown on gelatin-coated TC-treated glass chamber slides (BD Falcon) and incubated in starvation medium [Endothelial Cell Basal Medium 2 supplemented with 2.5% FBS] for 2 h before stimulation with VEGF-B$_{167}$, VEGF-B$_{186}$ (both 100 ng/ml) or vehicle (Control) in Endothelial Cell Basal Medium 2 supplemented with 1% FBS for 10, 20, 30 min, or 2 h in the presence of native human low-density lipoprotein:Dil (Dil-LDL) (10 min: 20 μg/ml, 20 and 30 min: 5 μg/ml, 2 h: 200 ng/ml). To specifically evaluate LDL binding to the cell surface, HUVECs were pre-stimulated with VEGF-B$_{186}$ (100 ng/ml) or vehicle (Control) for 10, 20, or 30 min at 37°C and thereafter the cells were put on ice before adding the Dil-LDL compound for an additional 10 min on ice. Cells were thereafter washed with PBS and fixed with 4% PFA. Nuclei were stained with DAPI, and coverslips were mounted. Three slides per condition were analyzed with microscopy as described above for the glucose uptake assay.

**TopFluor-cholesterol uptake assay**

HUVECs were grown on gelatin-coated glass coverslips and incubated in starvation medium [Endothelial Cell Basal Medium 2 supplemented with 2.5% FBS] for 2 h before stimulation with VEGF-B$_{167}$, VEGF-B$_{186}$ (both 100 ng/ml), or vehicle (Control) for

20 min or 2 h in the presence of 2.5 μM TopFluor-cholesterol. Cells were washed 3× with PBS and fixed for 15 min with 4% PFA followed by mounting with ProlongGold mounting medium. Three coverslips per condition were evaluated and imaged with a Zeiss Observer Z.1 microscope as described earlier for the glucose uptake assay.

**Filipin staining**

Cells or fresh frozen tissue sections were post-fixed in 4% PFA and blocked for 15 min with PBS-100 mM glycine, followed by incubation with filipin (1:200 in PBS, stock: 12.5 mg/ml in ethanol) for 1 h shaking. Cells/sections were thereafter washed 2× with PBS, mounted and subjected to evaluation and image capture using a Zeiss Observer Z.1 microscope as described earlier for the glucose uptake assay. Each condition was performed in triplicate, and 15 pictures in total were taken per condition. Tissue sections were analyzed in duplicate (10 pictures per condition).

**Cholesterol quantification**

Equal amounts of HUVECs were grown in 10 cm dishes. One dish was used per condition, and conditions were measured in duplicate. After incubation in starvation medium [Endothelial Cell Basal Medium 2 supplemented with 2.5% FBS] for 2 h, cells were stimulated with VEGF-B$_{186}$ (100 ng/ml) or vehicle (Control) for 20 min. Cells were thereafter washed with PBS and scraped into 100 μl PBS on ice. Cells were centrifuged at 1,500 g for 3 min at 4°C, and pellet was resuspended in 200 μl chloroform:isopropanol:Igepal (7:11:0.1) for 30 min in a sonication bath. Debris was pelleted by centrifugation at 16,000 g for 10 min at RT. Supernatant was collected and centrifuged at 16,000 g for 5 min. The chloroform phase was dried at 50°C, and residual organic solvent traces were removed by vacuum centrifugation for 1 h. The isolated lipids/chloroform phase was analyzed for non-esterified cholesterol with the colorimetric cholesterol detection kit from Abcam according to manufacturer's description.

**Cholesteryl ester quantification**

HUVECs were stimulated as described above (cholesterol quantification). Cells were thereafter washed and scraped into PBS. After centrifugation at 1,000 g for 5 min at 4°C the cell pellet was frozen at −20°C. Lipids were extracted and CE was analyzed by mass spectrometry using UFLC-MS-based lipidomics analysis performed at the Swedish Metabolomics Centre at the Swedish University of Agricultural Sciences.

**Immunofluorescence staining**

Cells grown on gelatin-coated TC-treated glass chamber slides were incubated in starvation medium [Endothelial Cell Basal Medium supplemented with 2.5% FBS] for 2 h before stimulation with VEGF-B$_{167}$, VEGF-B$_{186}$ (both 100 ng/ml) or vehicle (Control) for indicated time periods. Cells were washed with PBS and fixed with 4% PFA. Cells were permeabilized for 30 min with PBS containing 0.1% saponine and 0.2% BSA (BB) and incubated with primary antibodies in BB o/n at 4°C. Detergents were

omitted in the staining protocol when evaluating cell surface expression. Cells were thereafter washed with PBS-0.2% Tween20 (PBS-T) and incubated with fluorescent secondary antibody and if indicated with DAPI for 2 h in BB at RT. Cells were washed with PBS-T, PBS, and H$_2$O and mounted with Prolong Gold. Images were taken with a Zeiss laser scanning microscope (LSM) 700 equipped with a 20× (N/A = 0.8), 40× Oil (N/A = 1.3) or 63× Oil (N/A = 1.4) objective. Representative images shown are 2D renderings of 4 μm thick z-stacks.

Tissue sections were permeabilized for 1 h with PBS containing 1% BSA and 0.1% Triton X-100. Sections were thereafter incubated with primary antibodies in PBS containing 0.5% BSA and 0.05% Triton X-100 at 4°C, o/n. Sections were washed 3× with PBS and incubated for 2 h with secondary antibody and DAPI if indicated in PBS containing 0.5% BSA and 0.05% Triton X-100 at RT. Sections were washed 3× with PBS, mounted, and imaged as described above.

## Immunohistochemistry

Transverse sections, 5 μm thick, of paraffin embedded hearts from wild-type C57BL/6 mice of mixed sex ($n = 4$) were collected on Superfrost Plus slides (Thermo Fisher Scientific Inc.). After de-paraffinization and rehydration, heat-induced epitope retrieval in high pH buffer (K8004; Dako) for 1 h in a kitchen steamer was performed, followed by 10 min pre-treatment with 0.025% trypsin-EDTA (Gibco, Life Technologies) at RT. Endogenous peroxidase activity was quenched with 0.3% hydrogen peroxide/methanol for 30 min. Unspecific binding sites were blocked with serum-free protein block (X0909; Dako) for 1 h. Primary antibodies; rabbit anti-mGlut1 (1:200, 07-1401; Millipore) and goat anti-m/rCD31 (1:200, AF3628; R&D Systems) were applied on consecutive heart sections and incubated o/n at 4°C. After several washes in PBS-T, biotin-conjugated secondary antibodies were applied for 1 h RT in block buffer (1:250; Vector Laboratories). After washes in PBS-T, the Vectastain Elite ABC kit (Vector Laboratories) was applied according to the manufacturer's instructions. Finally, the DAB peroxidase substrate kit (Vector Laboratories) was utilized for visualization. Evaluation and image capture (20× and 63× objective) were done using a Zeiss Axio Observer Z1 inverted microscope and the ZEN 2009 software (Carl Zeiss Microimaging GmbH). Brightness and contrast settings were changed in Photoshop CS5 to generate final image and were applied equally to the entire image and within the same set of images. Representative images are shown.

## Advanced image analysis with ImageJ/Fiji software

### Analysis of low-density lipoprotein receptor clusters in HUVECs
False nuclear LDLR staining was removed by separating the channels of the immunofluorescence images. The nuclei channel was analyzed with "threshold", "count particle", "create mask", "select all nuclei", and "add to ROI manager". Then, the channel for LDLR staining was selected, "image overlay from ROI", "delete", "no selection", "convert to gray image", "set threshold", "analyze particles": 5 for 60× and 12 for 40×-Infinity (μm$^2$). Then, area (cluster size) was measured and mean area to number of nuclei was calculated.

### Analysis of vessel-associated cholesterol
Channels of the immunofluorescence images were separated. Podocalyxin (vessel marker) channel was selected, threshold was set and detected area was converted to mask and added to ROI manager. Then, the filipin channel was selected and overlaid from ROI. Signal in ROI area was measured. Mean integrated density was determined for 10 images per animal/condition and normalized to wild-type mice.

## siRNA transfection

HUVECs were grown in antibiotic-free medium for at least one passage before seeded in 6-well plates ($3.5 \times 10^5$ cells per well) and grown o/n. Cells were transfected in serum-free Opti-MEM medium with RNAiMAX reagent from Invitrogen according to the manufacturer's instructions. Cells were transfected with 40 nM siRNA for 4 h. Then, the same volume of growth medium + 5% FBS was added o/n. The next day, medium was changed to maintenance medium (Endothelial Cell Growth Medium 2) for 2 h. Cells were trypsinized, re-plated on 24-well plates and grown for another 24 h. Then, cells were stimulated with VEGF-B and used for different assays or lysed for RNA isolation or Western blot analysis.

## shRNA transfection and induction

MS1 cells were cultured to 40% confluency in a 96-well plate. Cells were transfected with 10 μl shRNA-construct containing lentivirus (stock = $1 \times 10^6$ IFU) + 90 μl medium in the presence of 0.8 μg/ml polybrene. The next day, the medium was replaced by medium with puromycin (10 μg/ml) and clones were selected and expanded. Knockdown efficiency was analyzed on RNA level by qPCR or if possible on protein level by Western blot. Inducible (lacO)-shRNA-construct-containing cell lines were induced with 500 μM isopropyl β-D-1-thiogalactopyranoside (IPTG) for 6 days. On day 6, cells were stimulated with VEGF-B and further analyzed in different uptake assays or by Western blot for knockdown efficiency.

## Cell surface protein biotinylation

Cells in 10 cm dishes were incubated in starvation medium [Endothelial Cell Basal Medium 2 supplemented with 2.5% FBS] for 2 h before VEGF-B (100 ng/ml) or vehicle (Control) stimulation for indicated time periods. After 3× PBS wash, cells were incubated for 30 min with 1 ml Biotin solution (0.5 mg/ml in PBS) at 4°C shaking. Cells were thereafter washed 2× with PBS-100 mM Glycine and lysed in 1.5 ml lysis buffer [50 mM Tris-HCl pH 7.5, 150 mM NaCl, 1% Ipegal, 1% Triton X-100, and phosphatase inhibitor] for 30 min at 4°C shaking. Lysates were cleared by centrifugation at 16,000 $g$ for 10 min at 4°C. Supernatant was transferred to a new tube, and 50 μl 50% streptavidin-sepharose slurry was added. Samples were rotated at 4°C o/n. Supernatant was collected and the sepharose beads were washed 3× with TBS [50 mM Tris-HCl, pH 7.6; 150 mM NaCl] and protein was released by boiling with 50 μl 3× SDS sample buffer [1×: 65 mM Tris-HCl pH 6.8, 10% glycerol, 2% SDS]. Samples were analyzed by SDS-polyacrylamide-gel electrophoresis (SDS-PAGE, Thermo Fischer Scientific) and Western blot.

## Cell fractionation by sucrose density gradient centrifugation

HUVECs in 10-cm dishes were incubated in starvation medium [Endothelial Cell Basal Medium 2 supplemented with 2.5% FBS] for 2 h before VEGF-B (100 ng/ml) or vehicle (Control) stimulation for indicated time periods. Three dishes were used per condition. Cells were washed on ice 2× with 10 mM HEPES pH 7.4 and scraped into 600 μl sucrose solution [15% sucrose, 10 mM HEPES pH 7.4, protease inhibitor, and phosphatase inhibitor]. Cells were homogenized by squeezing 7× through a 0.9 × 40 mm needle and 3× through a 0.4 × 40 mm needle. Samples were centrifuged at 1,000 ×$g$ for 5 min at 4°C. Samples were thereafter added on top of a stepwise sucrose gradient and centrifuged in an ultracentrifuge equipped with a SW41Ti rotor at 100,000 $g$ for 18 h at 4°C. Fractions of 500 μl were collected from the top: S1-4, 15% sucrose; S5, 20%; S6, 25%; S7, 27%; S8+9, 30%; S10+11, 35%; S12+13, 38%; S15+16, 40%; S17, 45%; S18, 48%; S19, 50%; S20, 52%; S21, 55%; S22, 60%; and S23, 70% sucrose. Samples were mixed with 6× SDS sample buffer, and 40 μl were analyzed by SDS-PAGE and Western blot.

## GLUT1 and LDLR protein expression analysis

For GLUT1 and LDLR protein quantification, HUVECs were grown in 10-cm dishes and incubated in starvation medium [Endothelial Cell Basal Medium 2 supplemented with 2.5% FBS] for 2 h before VEGF-B (100 ng/ml) or vehicle (Control) stimulation for 15 min or 2 h. Three to 4 dishes were used per condition. Protein samples were obtained by direct lysis in Laemmli buffer with β-mercaptoethanol, boiled, and separated with SDS-PAGE and thereafter transferred onto a nitrocellulose membrane (Thermo Fischer Scientific). Detection of GLUT1 and LDLR proteins was done using specific rabbit anti-GLUT1 (Millipore) and LDLR antibodies (20R-LR002; Fitzgerald) and mouse anti-β-actin antibodies (AB6276; Abcam) were applied as loading control.

## Endothelial cell isolation

ECs were isolated from lungs from C57BL/6 $Ldlr^{+/+}$ and $Ldlr^{-/-}$ pups (P4-P10). Prior to dissection, lungs were inflated with 1 mg/ml collagenase type I (Roche) + DNase I (20 U/ml) in PBS through the trachea. Lungs were collected, cut into pieces, and digested in 1 mg/ml collagenase A + 20 U/ml DNase I in PBS for 1.5 h at 37°C. Then, cells were mixed 1:1 with Endothelial Cell Growth Medium MV2 containing both antibiotics and antimycotics and filtered through a 70 μm cell strainer and collected by centrifugation. The pellet was washed 2× with PBS and resuspended in PBS-1% BSA. Suspension was added to Dynabeads Biotin Binders (Invitrogen) coupled with biotinylated rat anti-mouse Pecam-1 antibody (BD #553371) for 1 h at 4°C. Coupled beads were washed 8× with PBS-1% BSA, 1× with PBS, and 1× with Endothelial Cell Growth Medium MV2. Cells were resuspended in Endothelial Cell Growth Medium MV2 and plated in gelatin-coated 24-well plates. Cells from animals with the same genotype were pooled after 2-7 days and re-plated for VEGF-B stimulation and cholesterol staining.

## RNA isolation, cDNA synthesis, and quantitative real-time PCR

Total RNA was isolated from hearts from 20-week-old male C57BL/6 $Vegfb^{+/+}$ ($n = 6$), $Vegfb^{+/-}$ ($n = 6$) and $Vegfb^{-/-}$ ($n = 6$) mice using TRIzol extraction prior to isolation with the RNeasy Kit (Qiagen). RNA isolation from cultured cells was performed using the Qiagen RNeasy Kit, according to manufacturer's instructions. RNA concentration was determined with a NanoDrop instrument, and RNA was transcribed to cDNA with the iScript cDNA synthesis kit (Bio-Rad). Quantitative real-time PCR (qPCR) was performed using 10 ng cDNA and run in triplicates. cDNA was mixed with SYBR Mix and 250 nM primer and $H_2O$ to 20 μl. Samples were run in a Rotor-Gene Q from Qiagen or Corbett Research for 40 cycles (1× 95°C 3 min, 40× 3 s 95°C, 20 s 59°C, 2 s 72°C). Samples were normalized to *ribosomal protein L19* (*RPL19*) expression.

## Microarray analysis

Endothelial cell isolation from hearts from age-matched (12 weeks old) C57BL/6 $Vegfb^{+/+}$ ($n = 4$) and $Vegfb^{-/-}$ ($n = 4$) male mice or from hearts from 20 w old C57BL/6 wild-type mice was performed as described before (Hagberg *et al* 2010). In brief, mice were anesthetized (Hypnorm/Dormicum) and perfused with Hepes buffer (HBSS, Invitrogen) supplemented with 1% BSA via the left ventricle. Thereafter, the heart was removed and chopped into small pieces and incubated with 5 mg/ml collagenase A, 0.7 mg/ml hyaluronidase (both Roche), and 100 U/ml DNase I (Invitrogen) containing solutions at 37°C to disintegrate the tissue pieces. For further dissociation and removal of larger tissue remnants, the tissue lysates were passed through first a 70 μm mesh followed by passage through a 40 μm mesh (Falcon). The cell suspensions were pelleted by centrifugation at 200 $g$ for 10 min. Then, the cell-pellets were resuspended in HBSS containing 1% BSA and Dynabeads biotin binder (Invitrogen), which had been pre-coupled with biotinylated rat anti-Pecam1 antibody (MEC13.3, BD Pharmingen), were added to the cell suspension and incubated for 60 min at 4°C. Endothelial cells bound to the Dynabeads were extracted from the cell suspension by adsorption to a magnetic particle collector (Invitrogen). After repeated washing steps, the endothelial cell pellet was lysed in RLT buffer for RNA isolation, using the RNeasy kit (Qiagen) according to the manufacturer's instructions. The obtained RNA concentration and quality were determined using a NanoDrop and Bioanalyzer, respectively. Total RNA was used for microarray hybridization using Affymetrix GeneChip® Mouse Gene 1.0 ST chips according to the manufacturer's instructions.

Correspondingly, total RNA extracted using standard protocols (RNAeasy kit, Qiagen) from HBMECs stimulated with VEGF-A$_{165}$, PlGF, VEGF-B$_{167}$, or VEGF-B$_{186}$ for 6 h ($n = 3$ cell dishes/condition) after pre-starvation for 2 h in Endothelial Cell Basal Medium MV2 supplemented with 2.5% FBS was subjected to microarray hybridization using Affymetrix GeneChip® Human Gene 1.0 ST chips.

Microarray hybridization and analysis were carried out at the Bioinformatics and Expression analysis Core Facility, Department of Biosciences and Nutrition, NOVUM, Karolinska Institutet, Huddinge, Sweden. The generated microarray signals were PLIER normalized using global median. The gene-expression analysis was done comparing the mean values for the $Vegfb^{+/+}$ and $Vegfb^{-/-}$ groups ($n = 4$), or the control treated cells with the VEGF-B$_{167}$- or VEGF-B$_{186}$-treated cells ($n = 3$), respectively.

## Seahorse® live-cell metabolic assay

HUVECs were seeded at a density of 20,000 cells per well in 0.1% gelatin pre-coated Seahorse® XF96 Cell Culture Microplates

(102416-100, Agilent) in Endothelial Cell Growth Medium 2. Growth medium was exchanged after 24 h. At confluency 48 h later, the cells were starved for 2 h in Endothelial Cell Basal Medium 2 supplemented with 2.5% FBS. Thereafter, the cells were stimulated with 100 ng/ml of VEGF-B$_{186}$ protein or vehicle (Control), $n$ = 43–44 wells/condition, for 1 h at 37°C, 5% CO$_2$. After that, the media was exchanged for glucose-free Seahorse XF DMEM Medium pH 7.4 (Agilent, 103575-100) containing 2.5% FBS and 100 ng/ml VEGF-B$_{186}$ and cells were stimulated for an additional hour at 37°C in a designated incubator for de-gassing and elimination of CO$_2$. Immediately following, the Seahorse® XF Real-Time ATP Rate Assay (Agilent, 103592-100) was performed using the XFe 96 sensor cartridge (Agilent, 102416-100) according to the manufacturer's instructions. First, basal values (-glucose) were recorded. Sequential additions of glucose (Agilent, 103577-100, 5 mM final concentration), oligo-mycin (Agilent,103592-100, 1.5375 μM final concentration), and rotenone + antimycin A (Agilent,103592-100, 0.5125 μM final concentration) to the cells were hereafter executed by automated injections, and oxygen consumption rate (OCR) and extracellular acidification rate (ECAR) were analyzed in basal (-glucose) and induced (+glucose) conditions. The data was normalized to cell number, determined by quantification of cell nuclei after Hoechst staining. Data from a representative experiment repeated twice are shown.

## PET analysis of cardiac glucose uptake and cholesterol quantification on short-term anti-VEGF-B-treated mice

Wild-type C57BL/6 male mice (3 w old; Charles River, Germany) were put on high-fat diet (60% calories from fat, D12492, Research Diets), maintained for 17 weeks. Mice were thereafter randomly assigned to a short pre-treatment (1 week, 3 injections in total) with neutralizing anti-VEGFB antibodies (2H10; 400 μg/ mouse) or isotype-matched control antibodies (Control; 400 μg/ mouse). One set of mice was subjected to positron emission tomography (PET) analysis ($n$ = 4+4) and another set ($n$ = 5+5) sacrificed for organ harvest and filipin staining on heart tissue as described above.

PET was performed on a Focus120 MicroPET (CTI Concorde Microsystems). PET data were processed with MicroPET Manager (CTI Concorde Microsystems) and evaluated using the Inveon Research Workplace software (Siemens Medical Solutions). Mice were anaesthetized with isoflurane (5% initially and 1.5% to main-tain anesthesia) and placed on the camera bed on a heating pad (37°C). 2-[$^{18}$F]fluoro-2-deoxy-D-glucose, $^{18}$F-DG, was synthesized using an automated synthesis module (Fastlab, General Electric Medical Systems AB) and had passed routine clinical quality controls according to European Pharmacopoeia, Ph. Eur. Mice received tail vein intravenous injections of 10 MBq $^{18}$F-DG in a volume of 200 μl. The 60 min PET data acquisition started at injec-tion. All PET data were normalized with respect to the injected dose of each mouse (% ID/g). The signal from heart was collected from a region of interest of ≈240 mm$^3$. For statistical evaluation of PET data, two-way ANOVA followed by Tukey's multiple comparison test was used (T = 0.9–60 min). Linear regression analysis was performed to assess differences in initial glucose uptake rate (T = 0.9 min to 6.5 min).

## Glycogen measurement in heart tissue

Glycogen was measured with the Glycogen Colorimetric Assay Kit (K646-100, Biovision). Briefly, hearts were carefully dissected, bled out, and snap-frozen with liquid nitrogen. Frozen tissue was weighed and mixed with hydrolysis buffer to yield 150 mg/ ml. Tissue was homogenized in a bead homogenizer and boiled for 10 min. Supernatant was collected and used for colorimetric measurement according to manufacturer's instruction. All samples were spiked with glycogen to be in the optimal detec-tion range for the assay, and values were later corrected in the analysis. Furthermore, glucose background was extracted for all samples.

## Animal strains, welfare, and high-fat diet (HFD) experiments

C57BL/6 *Vegfb*$^{-/-}$ and BKS.C57BL/6 *db/db Vegfb*$^{-/-}$ mixed back-ground mice have previously been described [17,20,27]. BKS *Lepr*$^{db/db}$ (*db/db*) mice were purchased from Jackson Laboratory. Age-matched male mice were used in all studies unless otherwise stated. For mouse HFD studies, male C57BL/6 wild-type mice purchased from Charles River were fed a diet consisting of 60% kcal from fat (D12492, Research Diets) for 25 weeks from 5 weeks of age and injected twice weekly with isotype control or 2H10 antibody (400 μg/mouse) for the last 20 weeks on HFD. CD1 *Flt1 lacZ* knock-in mice (*Flt1*$^{tm1Jrt}$, *Flt1*$^{+/lacz}$) were a kind gift from Dr. Janet Rossant and Dr. Guo-Hua Fong, Mount Sinai Hospital, Toronto, Canada [70]. *Ldlr*$^{-/-}$ *Apob*$^{100/100}$ (*Ldlr*$^{-/-}$) or *Ldlr*$^{+/+}$ *Apob*$^{100/100}$ (*Ldlr*$^{+/+}$) mice were bred from *Ldlr*$^{+/-}$ *Apob*$^{100/100}$ that were generated from *Ldlr*$^{-/-}$ *Apob*$^{100/100}$ *Mttp*$^{flox/flox}$ *Mx1-Cre* described in [71] and backcrossed to C57BL/6 resulting in > 96.5% C57BL/6, < 3.5% 129/SvJae. All animals, independent of diet, had *ad libitum* access to chow and water, and were housed in standard cages in an environment with 12-h light–12-h dark cycles. All animal work using mice was conducted in accordance with the Swedish Animal Welfare Board at the Karolinska Insti-tutet, Stockholm, Sweden.

## Statistical analysis

*P*-values were calculated using a two-tailed unpaired Student's *t*-test related to indicated control. Comparisons between three experimen-tal conditions/groups or more were analyzed with one-way ANOVA followed by Dunnett's, Tukey's, or Sidak's multiple comparison test to calculate significances between the groups. Statistical evaluation of qPCR data was performed using one-way ANOVA and Fisher's LSD test. $P < 0.05$ were considered significant and indicated with asterisks: *< 0.05, **< 0.01, ***< 0.001.

# Data availability

The full microarray data sets presented in this publication have been deposited to the GEO database [https://www.ncbi.nlm.nih.gov/ge o/query/acc.cgi?acc = GSE146109] and assigned the accession number GSE146109.

**Expanded View** for this article is available online.

## Acknowledgements

We gratefully acknowledge Sofia Wittgren, Annelie Falkevall, Annika Mehlem, Karin Franzén, Aránzazu Rossignoli, Agnieszka Martowicz, and the MBB animal facility for support for animal experiments, provision with animal tissue, and help with experimental protocols and reagents. We furthermore gratefully acknowledge Erik Samén and Li Lu at the PET imaging center at KI for evaluation of [18]F-DG PET data. We thank Pierre Scotney and Andrew Nash at CSL Ltd. (Melbourne, Australia) for providing the monoclonal anti-VEGF-B antibody 2H10, and the isotype-matched monoclonal control antibody. This work was supported by grants from the Swedish Research Council (2016-02593, 2017-01794), the Swedish Heart and Lung Foundation (20120077, 20150547), the Swedish Cancer Society (CAN 2016/633), Novo Nordisk Foundation (NNF16OC0021172, NNF17OC0026942), and Karolinska Institutet and CSL Ltd. (Melbourne, Australia). C.M. is a recipient of an EMBO Long-Term Fellowship co-funded by Marie Curie Actions. L.M is supported by the Swedish Society for Medical Research.

## Author contributions

CM and IN designed the study, conducted experiments, and wrote the manuscript. LM partly designed the study, conducted experiments, and revised the manuscript. MZ and BHS conducted experiments and revised the manuscript. JS contributed to experimental design and scientific discussion and revised the manuscript. UE initiated and partly designed the study, contributed to scientific discussion, and revised the manuscript.

## Conflict of interest

U.E. is a consultant to CSL Ltd. I.N. and U.E. are inventors on patents describing the role of VEGF-B in diabetes and diabetic complications. CM, IN, LM, MZ, JS, and UE are shareholders in a company in the diabetes field.

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
