## [Review Process File · EMBO Reports]

VEGF-B Signaling Impairs Endothelial Glucose Transcytosis by Decreasing Membrane Cholesterol Content

Christine Moessinger, Ingrid Nilsson, Lars Muhl, Manuel Zeitelhofer, Benjamin Heller Sahlgren, Josefin Skogsberg, and Ulf Eriksson

Review timeline:

Submission date:	24 September 2019
Editorial Decision:	2 October 2019
Revision received:	28 February 2020
Editorial Decision:	3 April 2020
Revision received:	7 April 2020
Accepted:	21 April 2020

Editor: Deniz Senyilmaz-Tiebe

Transaction Report:

1st Editorial Decision

2 October 2019

Thank you for submitting your manuscript entitled " VEGF-B Signaling Impairs Endothelial Glucose Transcytosis via an LDLR-dependent Decrease in Membrane Cholesterol Loading " to EMBO Reports. Your manuscript was previously reviewed at another journal. I have now looked at everything carefully.

As you can see, the referees express interest in the proposed role of VEGF-B signalling on endothelial glucose uptake. However, they also raise a number of concerns that need to be addressed to consider publication here. In particular,

- Referees argue that the effects of VEGF-B signalling on glucose uptake could be an indirect effect of changes in fatty acid uptake (referee #1, point 2 and referee #3, point 1). Referee #3 recommends measuring glycolysis fatty acid oxidation and ATP production at the time point selected for figure 4 to address this issue.
- Referees #2 (point 1) and #3 (point 2) would like to see if the increase in glucose uptake in response to VEGF-B is also translated into changes in glucose metabolism of endothelial cells.
- Referee #3 questions the proposed potential therapeutic application of VEGF-B to alleviate type II diabetes related symptoms (second part of point 2). If you can address this concern experimentally, it would strengthen the manuscript. If not, it is important to discuss this.
- It is important to address the caveats regarding the effects of LDLR knockdown on cholesterol levels (referee #1, point 7 and referee #3, point 4) and the effects of FLT1 and NRP1 depletion on glucose uptake (referee #1 point3).
- Referees point out that subcellular localization of Glut1 was not conclusively demonstrated (referee #1 point 4, referee #3 point2). Moreover, referee #3 requests stronger evidence supporting the changes in cholesterol levels and additional controls for LDLR staining (points 3 and 4).
- Referee #2 would like to have more insight into how VEGF-B would affect LDL recycling. Please discuss the possibilities if you cannot address this experimentally.
- Lastly, referee #2 recommends an unbiased transcriptomic approach to have a better view on the effects of VEGF-B signalling on endothelial metabolism. I think this would strengthen the

manuscript.

Given these constructive comments, we would like to invite you to revise your manuscript with the understanding that the referee concerns (as detailed above and in their reports) must be fully addressed and their suggestions taken on board. Please address all referee concerns in a complete point-by-point response. Acceptance of the manuscript will depend on a positive outcome of a second round of review. It is EMBO reports policy to allow a single round of revision only and acceptance or rejection of the manuscript will therefore depend on the completeness of your responses included in the next, final version of the manuscript.

I look forward to seeing a revised version of your manuscript when it is ready. Please let me know if you have questions or comments regarding the revision.

REFEREE REPORTS

Referee #1

This study follows the Eriksson lab past work regarding the role of VEGF-B signaling on endothelial cells, this time analyzing its ability to reduce endothelial glucose uptake/transport and membrane cholesterol levels, through possibly an LDLR-mediated mechanism. The manuscript provides intriguing data, but there are several concerns that must be addressed in order to strengthen this manuscript, shed light on some confusing results, and fortify their conclusions. Overall the conclusions are tantalizing, but somewhat premature, and need significant strengthening.

1. In Figure 1B, the effect of B186 in reducing glucose transport gradually wore off over time - why is this the case? There should be some more discussion about this.
2. Figure 1A and related supplemental figures neatly demonstrate that VEGF-B signaling decreases glucose uptake in vitro, but the data in C, D, E are insufficient to convince that the same is occurring in vivo. Previous work from this lab has shown that VEGF-B signaling is required for fatty acid uptake. The heart's increased glucose uptake may just be because there are fewer lipids available to use as fuel, and not necessarily a loss of some VEGF-B specific signaling pathway inhibiting endothelial glucose transport. This conclusion is overinterpreted.
3. Figure 2D attempts to communicate the point that knockdown of FLT1 and NRP1 ablates VEGF-B mediated glucose uptake reduction. However, even without B186, those knockdowns reduce glucose uptake by almost as much as the B186 does in control siRNA cells. Why does knocking down receptors responsive VEGF-B signaling reduce glucose uptake if VEGF-B signaling itself reduces that very same uptake? It is paradoxical and muddles the conclusion that removing FLT1 and NRP1 prevents VEGF-B mediated reduction of glucose uptake.
4. The data in Figure 3 are not sufficient to make conclusions about the subcellular localization and movement of Glut1 in response to B186. Particularly in 3C, there is a great deal of variability and more N than just 2 for control and 2 for B186 must be used to make any concrete statements about Glut1's distribution.
5. Figure 4G shows that there is no shift in the type of cholesterol esters present after VEGF-B stimulation, but is there a difference in the total number of esters? This is the conclusion the authors make in the text, but the data only show relative numbers.
6. Figure 5 has one minor error: the units for cholesterol in B are given in mg/mL when it actually should be $\mu\text{g/mL}$, as according to the figure legend.
7. In Figure 6B, why doesn't cholesterol as measured by Filipin decrease under even the control conditions in LDLR1 knockdown cells? The knockout mouse ECs certainly have lower cholesterol, as shown in 6D. This is to be expected, considering LDLR's role in cholesterol homeostasis. So why does knocking this gene down in vitro not reduce cholesterol?
8. On a related note: if VEGF-B is reducing glucose uptake via impaired LDLR recycling (as the

authors posit), then knockdown of LDLR1 alone should reduce glucose uptake - and yet that result is not observed in 6C. Why is that?

9. Figure 7 has the same issue as 1C/D/E - that is, the effects seen in the VEGF-B knockout animals are not conclusively due to impaired VEGF-B signaling related to endothelial glucose uptake; rather, loss of VEGF-B in every tissue may be leading to unrelated compensatory mechanisms due to the decreased abundance of fatty acids.

Referee #2

Summary:

In the manuscript "VEGF-B signaling impairs endothelial glucose transcytosis via an LDLR-dependent decrease in membrane cholesterol loading", Moessinger and colleagues investigate VEGF-B regulated nutrient transport processes in the endothelium. The work is based on previous studies by the laboratory, which demonstrated that VEGF-B regulates fatty acid uptake and transcytosis in muscle tissues and that VEGF-B inhibition correlates with improved glucose handling in various disease models. Here the authors report that VEGF-B decreases endothelial glucose transport by altering cholesterol composition of the plasma membrane. This mechanism involves changes in membrane cholesterol loading and LDL receptor recycling, which is impaired in response to VEGF-B stimulation.

General Comments:

The mechanisms that regulate endothelial nutrient transport processes are still insufficiently characterized. Understanding these processes is, however, of central importance because proper nutrient delivery is one of the ancestral functions of the vasculature whose deregulation can cause disease. The paper by Moessinger and colleagues provides new insights into this area by describing a novel VEGF-B-dependent mechanism that causes changes in glucose uptake and transcytosis in endothelial cells.

Overall, the paper is well structured, written, and discussed. The data are of good quality and are also nicely presented. The findings presented by the authors refine our understanding of VEGF-B signaling and should be of interest for researchers in different fields (e.g., vascular biology, metabolism, diabetes, lipids).

I only have some minor comments the authors might want to address in a revised version of the manuscript.

Specific Comments - Minor:

1. The authors show that VEGF-B alters glucose uptake and transcytosis in endothelial cells but did not assess whether VEGF-B has direct effects on endothelial glucose metabolism. Since endothelial cells heavily rely on glucose, it would be important to know if VEGF-B stimulation affects glycolysis, glucose oxidation, or overall mitochondrial activity. In vitro studies suffice to address this point.
2. The authors should also perform an unbiased transcriptomic analysis in endothelial cells to analyze the impact of VEGF-B on the expression of nutrient transporters and metabolic enzymes. Excluding major changes in the metabolic machinery would provide further evidence for the suggested model.
3. How does VEGF-B affect LDL receptor recycling in the endothelium? Any idea how signaling through VEGFR1 and NRP1 is molecularly linked to the recycling machinery?

Referee #3

This paper describes a unique model by which the paracrine release of VEGFB from underlying parenchyma activates VEGFB signaling in the endothelium to decrease glucose uptake via a LDLR dependent decrease in membrane cholesterol. In addition to a reduction in glucose uptake, VEGFB signaling also increase fatty acid uptake, but the authors speculate that this increase is not dependent on a reduction in glucose uptake. Although innovative, several key experiments are missing to fully support the findings.

I have the following concerns:

General concept issues:

1. The authors suggest that the mechanism of an increase fatty acid uptake and decrease in glucose uptake in response to VEGFB signaling are not related. However, the evidence to support these claims have not been fully addressed. Likely the Randle Cycle is probably not governing these effects, as endothelial cells overwhelmingly favor glucose for energy production by way of anaerobic glycolysis (>90%) compared to other energy producing pathways, whereas for the Randle Cycle to be fully "operational" glucose must be oxidized in the mitochondria to inhibit fatty acid oxidation via malonyl CoA generation. Alternatively, according the Randle cycle, fatty acid oxidation leads to inhibition of pyruvate dehydrogenase (PDH) activity, which reduces glucose oxidation, which again, may not be as relevant in endothelial cells given the reliance on anaerobic glycolysis. Although the Randle cycle may not be governing the observations herein, demonstrating that VEGFB activation reduces glucose uptake prior to an increase in fatty acid uptake, does not definitively rule out that the decrease in glucose uptake leads to an increase in fatty acid uptake and utilization. Thus, measuring glycolysis, fatty acid oxidation, and ATP production at the time-point selected for Figure 4 is necessary to clarify these issues.

2. The authors show several experiments in which VEGFB stimulation reduces glucose uptake, whereas VEGFB knockout increases glucose uptake in vivo (18F-FDG). It would be useful to determine whether or not this increase in glucose uptake is translated to an increased metabolism of the substrate; i.e., glycolysis and glucose oxidation should be measured to support the claims that the increased rate of glucose uptake is primarily used for transcytosis to underlying tissue. Along these lines, the authors speculate that VEGFB neutralizing antibodies may be beneficial in preventing or treating conditions of type II diabetes, as these therapies would maintain glucose uptake and prevent elevated fatty acid uptake and potential lipotoxicity in underlying parenchyma. However, several lines of evidence suggest that excessive glucose uptake within endothelial in the setting of hyperglycemia is particularly harmful considering that glucose uptake in endothelial cells is not limited by insulin signaling as the authors have highlighted. Therefore, VEGFB antagonism in the face of hyperglycemia would further increase endothelial glucose uptake, which may lead to vascular dysfunction. Proper experiments to rule out a potential vascular toxicity with VEGFB antagonism are needed, particularly in the setting of elevated blood glucose.

Specific experimental issues:

1. In Fig 1B, how was transcytosis versus glucose metabolism and paracellular movement scored in this transwell assay?

2. The GLUT1 imaging looks very non-specific and should be validated using siRNA to GLUT1. The separation techniques used for GLUT1, NRP1 and VEGFR1 are less than optimal since the authors cannot separate ER from PM. This is not surprising since most sub cellular techniques for these compartments do not work well in endothelial cells. Also, markers for subcellular compartments are critical for the interpretation of Fig 3C. It is suggested that surface biotinylation of GLUT1 is assessed as is done for LDLR in Fig 6.

3. The mechanism surrounding VEGFB regulation of plasma membrane cholesterol content and LDLR is interesting, but of some concern. Firstly, the levels of free cholesterol (FC) are measured using the antibiotic filipin that binds FC, not CE. Filipin is notorious for rapidly quenching on slides and tissues, making it extremely difficult for comparative, quantitative measurements. The pictures of the cells are not convincing, and the images are much less convincing in tissues. Based on filipin staining, the VEGF-B treated cells lose 50% of the FC, and this should be either be transported out via ABC transporters or esterified. The one experiment where cholesterol is measured is Fig 4E but this is not cholesterol from the PM, but from the cells. Where is 20% of the cholesterol going since it cannot be metabolized?

In Fig 5, direct measurements of PM cholesterol are needed. An additional control for mbCD would be the uptake of an additional tracer, such as albumin or dextran.

4. It is interesting that knockdown of LDLR reduces VEGFB mediated reductions in glucose uptake. Western blotting for LDLR is needed in the KD and VEGFB treated cells. Since LDLR KD

attenuates VEGFB-mediated decrease in cholesterol content and glucose uptake, yet siLDLR alone did not significantly decrease cholesterol/uptake by itself. If the mechanism is via recycling of the LDLR, then the si LDLR treated group should be shifted down and not be affected by the VEGFB inhibitor. Controls are needed for LDLR IF, since these Ab are notorious for non-specificity.

5. In 6D, VEGFB suppression of glucose uptake and the levels of NRP and FLT and another sterol responsive gene (HMGCoaR) are critical for interpretation of the experiments. Again, chemical measurements of FC and CE would strengthen this data.

6. There are no control plasma membrane markers in Fig 6F, making this impossible to interpret.

7. If VEGF-B impacts cholesterol, then ratiometric imaging of membrane fluidity should be assessed.

1st Revision - authors' response

28 February 2020

EMBOR-2019-49343-T Response to reviewers

Referee #1

This study follows the Eriksson lab past work regarding the role of VEGF-B signaling on endothelial cells, this time analyzing its ability to reduce endothelial glucose uptake/transport and membrane cholesterol levels, through possibly an LDLR-mediated mechanism. The manuscript provides intriguing data, but there are several concerns that must be addressed in order to strengthen this manuscript, shed light on some confusing results, and fortify their conclusions. Overall the conclusions are tantalizing, but somewhat premature, and need significant strengthening.

We appreciate the overall positive response to our work and hope that remaining issues have been resolved in the revised manuscript.

1. In Figure 1B, the effect of B186 in reducing glucose transport gradually wore off over time - why is this the case? There should be some more discussion about this.

We believe that showing that VEGF-B is not only affecting uptake, but also endothelial transcytosis of glucose is of vital importance for the overall conclusions of the manuscript.

In this experiment, the cumulative effect on glucose transcytosis over time is measured and the raw data shows an increased signal for 2-NBDG over time when sampled from the lower chamber in both Control and VEGF-B settings. Of note, in the presentation of the data, Controls had been set to 1 for each respective timepoint to make the comparison to VEGF-B treatment clearer (see figure below). With time, the difference between VEGF-B treated and Control treated cells became less apparent as pointed out by the reviewer.

The setup of the kinetic transwell assay is intricate as it requires maintenance of a functional glucose gradient over the endothelial monolayer to enable unidirectional transport throughout the experiment. In addition, cell media is replaced for Krebs-Ringer solution for an extensive period of time (2 hours) which may influence cell behaviour and responsiveness to VEGF-B. In our hands, 2-NBD-glucose tracer net uptake was stable and did not reach saturation in ECs during a 40 min uptake session (Fig EV1E (former Supplementary Fig 1)), signifying the appropriateness of the 20 min and 30 min incubation times in the kinetic transwell experiment. All other 2-NBD-glucose experiments in the study has been evaluated using a 20 min long exposure time for 2-NBD-glucose. Thus, for the longer evaluation timepoints (>45 min), effects related to Krebs-Ringer starvation and/or partial reversal of glucose transport direction that may be differentially regulated by VEGF-B (i.e. re-uptake of 2-NBD-glucose from the lower chamber) are important issues that we have not controlled for.

Collectively, of concern for prolonged exposure to Krebs-Ringer solution and to ensure that only unidirectional transport is being measured, we have therefore decided to replace the kinetic setup

(upper panel below) with data from the fixed exposure time of 20 min with 2-NBDG after 2 h stimulation with VEGF-B186 (lower panel below), conforming with all other evaluations using the 2-NBDG tracer in the manuscript. Materials and Methods have been revised accordingly.

2. Figure 1A and related supplemental figures neatly demonstrate that VEGF-B signaling decreases glucose uptake in vitro, but the data in C, D, E are insufficient to convince that the same is occurring in vivo. Previous work from this lab has shown that VEGF-B signaling is required for fatty acid uptake. The heart's increased glucose uptake may just be because there are fewer lipids available to use as fuel, and not necessarily a loss of some VEGF-B specific signaling pathway inhibiting endothelial glucose transport. This conclusion is overinterpreted.

We believe that VEGF-B is not only affecting endothelial cell uptake, but also transcytosis of glucose to underlying tissue, of vital importance for the overall conclusions of the manuscript.

The properties of the cardiac vasculature, comprising a continuous and low permeable monolayer of endothelial cells, proposes that the endothelial cells constitute a barrier for tissue glucose delivery and that endothelial cells consequently may influence glucose utilization in heart. VEGF-B is highly expressed in cardiomyocytes *in vivo* and signals via its receptors VEGFR1 and NRP1, exclusively co-expressed in cardiac endothelial cells, suggesting a molecular crosstalk between cardiomyocytes and endothelial cells in directing tissue nutrient uptake (Fig 2F and [1]). Along these lines, an altered need of energy supply may initiate tissue-derived responses affecting endothelial transcytosis of energy substrates from blood to the underlying tissue. Indeed, expression of VEGF-B is controlled by the metabolic sensor PGC1 α and cardiac VEGF-B levels has accordingly been shown to be significantly increased in diabetic mouse models [2]. Together, these aspects lend support to the evaluation of glucose uptake in relation to VEGF-B in Fig 1.

Vegfb^{-/-} mice exhibit a phenotype composed of decreased cardiac FA uptake and increased cardiac glucose uptake compared to wildtype control mice, suggesting that VEGF-B controls a nutrient substrate switch in the heart [3,4]. To exclude confounding effects coupled to unknown developmental or metabolic compensations in the congenital Vegfb^{-/-} mouse, we chose to evaluate glucose uptake and accumulation after acute pharmacological inhibition of VEGF-B using dynamic ¹⁸F-DG PET imaging *in vivo* (Fig 1D, E). A dynamic, in contrast to a steady-state, PET acquisition allowed us to specifically assess temporal differences in glucose uptake rate in contrast to potential indirect effects on glucose metabolism that may contribute to the signal in the latter end of the acquisition time-frame (Fig 1E). The significantly increased rate of glucose uptake observed during the first couple of minutes after glucose tracer injection (Fig 1E) proposes that the effect of VEGF-B targeting on overall cardiac glucose uptake is related to an enhanced rate of endothelial uptake and transcytosis, representing the first barrier through which glucose must pass on its way from blood to the underlying tissue.

Although ¹⁸F-DG PET acquisition *in vivo* does not distinguish between endothelial cell and cardiomyocyte glucose uptake, cardiac glycogen content complements the PET analysis and proposes that an increase in glucose uptake into the heart is made available to cardiomyocytes and stored as glycogen (Fig 1C). Transcriptional analysis on hearts from Vegfb^{-/-} mice and controls showed no significant differences in gene expression of rate limiting key enzymes involved in glycogen synthesis or degradation, suggesting that the increase in glycogen most probably arise from increased glucose uptake rather than changes in glycogen synthesis capacity (Fig EV1F). That the glucose is not utilized directly for ATP production in the heart, and instead is stored as glycogen, indicates that glucose is not a limiting factor, or preferred substrate, in cardiac tissue energy consumption in conditions of standard mouse housing.

To further exclude that the effect on impaired endothelial glucose transcytosis is indirect to more FAs available for endothelial intermediary metabolism, Seahorse experiments have now been performed and included in Fig EV3 (see detailed description to Reviewer 2, Q2). In summary, VEGF-B stimulation for 2 h does not change endothelial energy substrate use or energy balance, as measured by oxygen consumption rate (OCR) and extracellular acidification rate (ECAR). Comparable oxidative (mitochondrial) vs. glycolytic ATP production rates in both absence or presence of glucose stimulation further points to that endothelial cells utilize glucose as preferred energy substrate also in conditions of active VEGF-B signaling and overall decreased glucose uptake. These observations support a role of paracrine VEGF-B signalling in energy substrate partitioning at the level of the vascular interphase and transcytosis of energy substrates to the VEGF-B expressing tissue cells.

3. Figure 2D attempts to communicate the point that knockdown of FLT1 and NRP1 ablates VEGF-B mediated glucose uptake reduction. However, even without B186, those knockdowns reduce glucose uptake by almost as much as the B186 does in control siRNA cells. Why does knocking down receptors responsive VEGF-B signaling reduce glucose uptake if VEGF-B signaling itself reduces that very same uptake? It is paradoxical and muddles the conclusion that removing FLT1 and NRP1 prevents VEGF-B mediated reduction of glucose uptake.

We agree that interpretation of knock-down experiments should be done carefully.

We and others have previously identified that VEGF-B needs to engage with both of its receptors co-expressed in endothelial cells to elicit a biological response, and this was the rationale behind the knock-down experiments [3,5]. Notably, this dual dependence between VEGFR1 (FLT1) and NRP1 in VEGF-B mediated signaling has been characterized *in vivo* using adenoviral approaches and endothelial cell specific Nrp1 knock-out mice [3], and in Flt1 heterozygous mice, respectively [6].

It has been shown that knocking down a protein normally localized to the plasma membrane can influence the plasma membrane distribution of several other signaling proteins. For example, knock-down of the endocytic receptor low-density lipoprotein receptor-related protein 1 (LRP1) decrease the plasma membrane localization/expression of other proteins, here among NRP1 [7]. Of note, ligand stimulation of VEGF receptors leads to endocytosis and decreased plasma membrane localization of the receptor protein as well, thus in a way mimicking the effect of knocking down the receptor in the absence of ligand stimulation. The effect on glucose uptake in FLT1 or NRP1 knock-down endothelial cells may therefore potentially be indirect to unknown effects on glucose uptake in the plasma membrane, potentially related to decreased localization of either of these receptors in the plasma membrane. This raises an important but understudied issue with relevance for all types of knock-down/knock-out experiments *in vitro* as well as *in vivo*. The knock-down experiments in our study have been sufficiently repeated to provide representative data.

Notably, the response to VEGF-B was not hampered by a lower basal glucose uptake in endothelial cells. Treatment with phorbol 12-myristate 13-acetate (targeting PKC signaling) correspondingly reduced basal glucose uptake to a similar degree as VEGF-B signaling alone, albeit the cells retained their responsiveness to a further decrease in glucose uptake following co-treatment with VEGF-B (Fig EV2C).

Importantly, knock-down of KDR did not affect basal glucose uptake in endothelial cells. Stimulation with its ligand VEGF-A correspondingly did not affect glucose uptake either, pointing to a biologically significant differential and VEGFR1/NRP1 dependent effect on endothelial glucose transport (Fig 2D).

4. The data in Figure 3 are not sufficient to make conclusions about the subcellular localization and movement of Glut1 in response to B186. Particularly in 3C, there is a great deal of variability and more N than just 2 for control and 2 for B186 must be used to make any concrete statements about Glut1's distribution.

GLUT1 is situated in the plasma membrane and remains in the plasma membrane in response to VEGF-B stimulation, judged by both immunofluorescence (IF) stainings and sucrose gradient experiments (Fig 3 and Appendix Fig S2). GLUT1 is thus not stored intracellularly, in contrast to e.g. GLUT4. We have now improved the IF staining protocol and updated the panels showing GLUT1 localization in the EC plasma membrane as opposed to in the intracellular compartment in both Control and VEGF-B stimulated cells (Fig 3A). We have also included quantitative Western blot data showing that the total cellular GLUT1 protein pool was not changed after VEGF-B stimulation up to 2 h (Fig 3B).

A standard method to assess the lipid environment of proteins in cellular membrane is by isolation and separation in sucrose density gradients and co-localization with markers for different subcellular compartments. The challenge consists in that membranes comprise a very variable group of membrane subdomains including caveolae and membrane areas rich in cholesterol and sphingolipids of different ratios [8]. The subcellular spatial localization of GLUT1 has been investigated in HeLa cells using super-resolution imaging [9]. In their paper, GLUT1 distribution in the plasma membrane showed preferred localization to lipid rafts (i.e. cholesterol-rich microdomains) [9]. Intriguingly, treatment with methyl-beta-cyclodextrin (MβCD) reduced clustering of GLUT1 in the lipid raft microdomains [9], in line with our data from MβCD treatment or VEGF-B stimulation leading to reduced free cholesterol loading of plasma membrane, affecting GLUT1 mediated glucose uptake. This reference has now been included in the Discussion in support of our data.

5. Figure 4G shows that there is no shift in the type of cholesterol esters present after VEGF-B stimulation, but is there a difference in the total number of esters? This is the conclusion the authors make in the text, but the data only show relative numbers.

The lipidomic data provided quantitative measurements of the peaks representing the four most common cholesteryl ester (CE) species in Control vs. VEGF-B stimulated cells. We have from this data expressed the CE content as %. There might be additional CE species that was not possible to detect by the mass spectrometry and we have therefore revised the text in the Results section from “No differences in CE levels were however detected following VEGF-B stimulation (Fig 4G)” to “No differences in the levels of the major CE species were however detected following VEGF-B stimulation (Fig 4G)”. In the Fig 4G legend, the text was changed from “Quantification of cholesteryl esters (CE)...” to: “Quantification of the major cholesteryl ester (CE) species...”.

6. Figure 5 has one minor error: the units for cholesterol in B are given in mg/mL when it actually should be $\mu\text{g/mL}$, as according to the figure legend.

We thank the reviewer for finding this error. The unit has now been changed to $\mu\text{g/mL}$.

7. In Figure 6B, why doesn't cholesterol as measured by Filipin decrease under even the control conditions in LDLR1 knockdown cells? The knockout mouse ECs certainly have lower cholesterol, as shown in 6D. This is to be expected, considering LDLR's role in cholesterol homeostasis. So why does knocking this gene down in vitro not reduce cholesterol?

LDLR expression after LDLR knock-down was evaluated at the mRNA level, approximately 50% reduced, and likely less than that at the protein level, meaning that residual LDLR protein exist in the knock-down experiments which can be enough to ensure proper basal cholesterol homeostasis. In the presence of VEGF-B however, impaired recycling of residual LDLR to the cell surface presumably will lead to a detectable decrease in cholesterol content in the knock-down experiments (Fig 6B). This is different from the *Ldlr* knock-out cells where there is no residual LDLR present to be recycled.

8. On a related note: if VEGF-B is reducing glucose uptake via impaired LDLR recycling (as the authors posit), then knockdown of LDLR1 alone should reduce glucose uptake - and yet that result is not observed in 6C. Why is that?

Related to Q7 above.

LDLR expression after LDLR knock-down was evaluated at the mRNA level, approximately 50% reduced, and likely less than that at the protein level, meaning that residual LDLR protein exist in the knock-down experiments which can be enough to ensure proper basal cholesterol homeostasis. We observed in Fig 5 that endothelial cell glucose uptake stays functional with cholesterol loading within an optimal window.

9. Figure 7 has the same issue as 1C/D/E - that is, the effects seen in the VEGF-B knockout animals are not conclusively due to impaired VEGF-B signaling related to endothelial glucose uptake; rather, loss of VEGF-B in every tissue may be leading to unrelated compensatory mechanisms due to the decreased abundance of fatty acids.

Related to Q2 above.

Our study focuses on the role of VEGF-B signaling in endothelial cells in partitioning of glucose and FA transcytosis. We were therefore curious to look in tissues that normally express significant amounts of VEGF-B, such as cardiac tissue.

Besides *Vegfb*^{-/-} mice, acute pharmacological inhibition and models of mild cardiac VEGF-B overexpression was also evaluated [2] (Fig 7D-F). As the reviewer points out, the relationship between VEGF-B expression, membrane cholesterol content and glucose uptake data in heart are however only correlative. Generation of cardiac tissue specific models of *Vegfb* deficiency or overexpression was not feasible to do in this period of time.

The cardiomyocytes of the heart are flexible when it comes to utilization of energy substrates whereas endothelial cells rely on glucose. That glucose is not utilized directly for ATP production in

the heart, and instead is stored as glycogen, indicates that glucose is not a limiting factor, or preferred substrate, in cardiac tissue energy consumption in conditions of standard mouse housing.

To further exclude that the effect on impaired endothelial glucose transcytosis is indirect to more FAs available for endothelial intermediary metabolism, Seahorse experiments have now been performed and included in Fig EV3 (see detailed description to Reviewer 2, Q2).

Referee #2

Summary:

In the manuscript "VEGF-B signaling impairs endothelial glucose transcytosis via an LDLR-dependent decrease in membrane cholesterol loading", Moessinger and colleagues investigate VEGF-B regulated nutrient transport processes in the endothelium. The work is based on previous studies by the laboratory, which demonstrated that VEGF-B regulates fatty acid uptake and transcytosis in muscle tissues and that VEGF-B inhibition correlates with improved glucose handling in various disease models. Here the authors report that VEGF-B decreases endothelial glucose transport by altering cholesterol composition of the plasma membrane. This mechanism involves changes in membrane cholesterol loading and LDL receptor recycling, which is impaired in response to VEGF-B stimulation.

General Comments:

The mechanisms that regulate endothelial nutrient transport processes are still insufficiently characterized. Understanding these processes is, however, of central importance because proper nutrient delivery is one of the ancestral functions of the vasculature whose deregulation can cause disease. The paper by Moessinger and colleagues provides new insights into this area by describing a novel VEGF-B-dependent mechanism that causes changes in glucose uptake and transcytosis in endothelial cells.

Overall, the paper is well structured, written, and discussed. The data are of good quality and are also nicely presented. The findings presented by the authors refine our understanding of VEGF-B signaling and should be of interest for researchers in different fields (e.g., vascular biology, metabolism, diabetes, lipids).

I only have some minor comments the authors might want to address in a revised version of the manuscript.

We appreciate the overall positive response to our work and hope that remaining issues have been resolved in the revised manuscript.

Specific Comments - Minor:

1. The authors show that VEGF-B alters glucose uptake and transcytosis in endothelial cells but did not assess whether VEGF-B has direct effects on endothelial glucose metabolism. Since endothelial cells heavily rely on glucose, it would be important to know if VEGF-B stimulation affects glycolysis, glucose oxidation, or overall mitochondrial activity. In vitro studies suffice to address this point.

We thank the reviewer for this relevant point and suggestion to look at whether VEGF-B directly influences endothelial glucose metabolism as a potential mechanism of action. These new data are now presented in Fig EV3. The manuscript text has been updated accordingly.

In summary, VEGF-B stimulation for 2 h does not change endothelial energy substrate use or energy balance, as measured by oxygen consumption rate (OCR) and extracellular acidification rate (ECAR). Comparable oxidative (mitochondrial) vs. glycolytic ATP production rates in both absence or presence of glucose stimulation further points to that endothelial cells utilize glucose as preferred energy substrate also in conditions of active VEGF-B signaling and overall decreased glucose uptake. These observations support a role of paracrine VEGF-B signalling in energy substrate partitioning at the level of the vascular interphase and transcytosis of energy substrates to the VEGF-B expressing tissue cells.

The following text is added to the Results section:

“To exclude the possibility that the observed decrease in glucose uptake in response to VEGF-B signaling is indirectly related to an altered cellular metabolism, the Seahorse® live-cell metabolic assay platform was utilized to assess energy substrate use and cellular energy balance. Intriguingly, HUVECs almost exclusively utilize glucose as energy substrate to produce ATP via glycolysis, both during starved and fed conditions (Fig EV3). Conversely, mitochondrial respiration and oxygen consumption were seemingly underutilized for ATP production purposes. Importantly, VEGF-B stimulation did not alter energy substrate utilization or total ATP production in the endothelial cells themselves, supporting a role of VEGF-B in energy substrate partitioning at the level of the vascular interphase and vascular transcytosis of energy substrates to underlying tissue cells.”

2. The authors should also perform an unbiased transcriptomic analysis in endothelial cells to analyze the impact of VEGF-B on the expression of nutrient transporters and metabolic enzymes. Excluding major changes in the metabolic machinery would provide further evidence for the suggested model.

We thank the reviewer for this relevant point and suggestion to look with an unbiased approach at whether VEGF-B directly influences expression of endothelial nutrient transporters and metabolic enzymes as a potential mechanism of action. These new data are now presented in Fig EV4 and the full sets of microarray data have been deposited in the GEO database and will be accessed via a separate link that will be provided in a few days. The manuscript text has been updated accordingly.

If VEGF-B is affecting overall endothelial cell metabolism, VEGF-B stimulation would lead to differential transcriptional regulation of rate-limiting proteins and enzymes. We have therefore performed an unbiased transcriptomic analysis of VEGF-B stimulated endothelial cells in order to look at this aspect. In summary, the data shows that there are, on the transcriptional level, no major changes in the metabolic machinery in the endothelial cells themselves in response to VEGF-B stimulation. These new data are presented in a heat map over selected genes involved in metabolism (Fig EV4). The selected genes also shown in Appendix Table S3. From this analysis, only three significantly differentially expressed (upregulated) genes by VEGF-B167, VEGF-B186 or both, were identified: CPT2, HK2 and SLC16A7. Although these gene products are known players in glucose and mitochondrial metabolism, they do not alone lend support to that a particular pathway in intermediary metabolism are regulated by VEGF-B in the time frame evaluated. Together with the Seahorse data (response to Q1) showing that VEGF-B stimulation for 2 h does not change endothelial energy substrate use or energy balance, we could conclude that VEGF-B signaling and effect on transcytosis in endothelial cells seems not directly consequential to endothelial intermediary metabolism.

The following text is added to the Results section:

...”Furthermore, unbiased transcriptional profiling of HBMECs stimulated with VEGF-B revealed no general effect on gene expression related to cell metabolism as summarized in Fig EV4. Collectively, these data suggest specific signaling driven by VEGF-B rather than a compensatory response to changed substrate availability and energy status as conferred by the Randle cycle [10].”

3. How does VEGF-B affect LDL receptor recycling in the endothelium? Any idea how signaling through VEGFR1 and NRP1 is molecularly linked to the recycling machinery?

These are outstanding and intriguing questions that we have yet not fully undertaken experimentally.

In the literature, LDLR recycling has been primarily investigated in the liver, but liver endothelial cells are heavily fenestrated and do not constitute a barrier for e.g. glucose transport, as does for example endothelial cells in the heart. In endothelial cells of e.g. cardiac tissue, the endocytosis/recycling machinery is most likely part of the mechanisms we are studying.

We have in the revised version of the manuscript included a more elaborate and thorough investigation of LDLR expression and LDL uptake in response to VEGF-B stimulation. In Fig 6G, the cell surface expression of the LDLR was assessed by immunostaining with N-terminal-specific antibody on non-permeabilized cells as a complement to the biotinylation experiment in Fig 6F. This confirmed that the LDLR localization at the cell surface is seemingly decreased in response to 10 min of VEGF-B stimulation. This correlated with decreased Dil-LDL binding to the endothelial

cells (evaluated at +4 °C to prevent endocytosis) as well as uptake of Dil-LDL (cells incubated in 37 °C) (Fig 6G).

Quantitative Western blotting revealed no difference in the total LDLR protein pool, suggesting an alternative mechanism to decreased cell surface localization than increased degradation (Appendix Fig S4A-B). The intracellular pool of the LDLR was investigated further in Appendix Fig S4C-E and S5. As the perinuclear accumulation of LDLR is less apparent in VEGF-B treated cells (Appendix Fig S4C-D), co-staining with Golgi (Appendix Fig S4E), endosomal (Appendix Fig S5A) and lysosomal (Appendix Fig S5B) markers were analyzed. No apparent changes in the co-localization of the LDLR with these cellular compartments were observed in response to VEGF-B treatment, other than the finding of a more dispersed intracellular distribution pattern in VEGF-B treated cells.

Regarding identification of a molecular link between VEGFR1/NRP1 signaling and LDLR recycling machinery is indeed very interesting and most likely involves the endocytosis/recycling machinery. Future studies involving high-resolution imaging are warranted to investigate this aspect in greater detail.

Referee #3

This paper describes a unique model by which the paracrine release of VEGFB from underlying parenchyma activates VEGFB signaling in the endothelium to decrease glucose uptake via a LDLR dependent decrease in membrane cholesterol. In addition to a reduction in glucose uptake, VEGFB signaling also increase fatty acid uptake, but the authors speculate that this increase is not dependent on a reduction in glucose uptake. Although innovative, several keys experiments are missing to fully support the findings.

We appreciate the overall positive response to our work and hope that remaining issues have been resolved in the revised manuscript.

I have the following concerns:

General concept issues:

1. The authors suggest that the mechanism of an increase fatty acid uptake and decrease in glucose uptake in response to VEGFB signaling are not related. However, the evidence to support these claims have not been fully addressed. Likely the Randle Cycle is probably not governing these effects, as endothelial cells overwhelmingly favor glucose for energy production by way of anaerobic glycolysis (>90%) compared to other energy producing pathways, whereas for the Randle Cycle to be fully "operational" glucose must be oxidized in the mitochondria to inhibit fatty acid oxidation via malonyl CoA generation. Alternatively, according the Randle cycle, fatty acid oxidation leads to inhibition of pyruvate dehydrogenase (PDH) activity, which reduces glucose oxidation, which again, may not be as relevant in endothelial cells given the reliance on anaerobic glycolysis. Although the Randle cycle may not be governing the observations herein, demonstrating that VEGFB activation reduces glucose uptake prior to an increase in fatty acid uptake, does not definitively rule out that the decrease in glucose uptake leads to an increase in fatty acid uptake and utilization. Thus, measuring glycolysis, fatty acid oxidation, and ATP production at the time-point selected for Figure 4 is necessary to clarify these issues.

We thank the reviewer for the insightful comments and excellent proposals. We also agree with the reviewer that the Randle Cycle in endothelial cells may not be relevant to endothelial selection of energy substrate transport to underlying tissue cells.

We have performed new experiments to evaluate whether intermediary metabolism in endothelial cells determines the basis for selecting substrates for transendothelial transport. We found no support of that VEGF-B stimulation would affect glucose utilization and shift in glucose metabolism for preferential mitochondrial oxidation and/or change overall ATP production. These new data are now presented in Fig EV3. The manuscript text has been updated accordingly. See also response to Reviewer #1.

In summary, VEGF-B stimulation for 2 h does not change endothelial energy substrate use or energy balance, as measured by oxygen consumption rate (OCR) and extracellular acidification rate (ECAR). Comparable oxidative (mitochondrial) vs. glycolytic ATP production rates in both absence or presence of glucose stimulation further points to that endothelial cells utilize glucose as preferred energy substrate also in conditions of active VEGF-B signaling and overall decreased glucose uptake. These observations support a role of paracrine VEGF-B signalling in energy substrate partitioning at the level of the vascular interphase and transcytosis of energy substrates to the VEGF-B expressing tissue cells.

The following text is added to the Results section:

“To exclude the possibility that the observed decrease in glucose uptake in response to VEGF-B signaling is indirectly related to an altered cellular metabolism, the Seahorse® live-cell metabolic assay platform was utilized to assess energy substrate use and cellular energy balance. Intriguingly, HUVECs almost exclusively utilize glucose as energy substrate to produce ATP via glycolysis, both during starved and fed conditions (Fig EV3). Conversely, mitochondrial respiration and oxygen consumption were seemingly underutilized for ATP production purposes. Importantly, VEGF-B stimulation did not alter energy substrate utilization or total ATP production in the endothelial cells themselves, supporting a role of VEGF-B in energy substrate partitioning at the level of the vascular interphase and vascular transcytosis of energy substrates to underlying tissue cells.”

2. The authors show several experiments in which VEGFB stimulation reduces glucose uptake, whereas VEGFB knockout increases glucose uptake in vivo (18F-FDG). It would be useful to determine whether or not this increase in glucose uptake is translated to an increased metabolism of the substrate; i.e., glycolysis and glucose oxidation should be measured to support the claims that the increased rate of glucose uptake is primarily used for transcytosis to underlying tissue. Along these lines, the authors speculate that VEGFB neutralizing antibodies may be beneficial in preventing or treating conditions of type II diabetes, as these therapies would maintain glucose uptake and prevent elevated fatty acid uptake and potential lipotoxicity in underlying parenchyma. However, several lines of evidence suggest that excessive glucose uptake within endothelial in the setting of hyperglycemia is particularly harmful considering that glucose uptake in endothelial cells is not limited by insulin signaling as the authors have highlighted. Therefore, VEGFB antagonism in the face of hyperglycemia would further increase endothelial glucose uptake, which may lead to vascular dysfunction. Proper experiments to rule out a potential vascular toxicity with VEGFB antagonism are needed, particularly in the setting of elevated blood glucose.

These are all very important questions raised by the reviewer. See also response to Reviewer #1, Q2

Although ¹⁸F-DG PET acquisition in vivo does not distinguish between endothelial cell and cardiomyocyte glucose uptake and accumulation, cardiac glycogen content complements the PET analysis and proposes that an increase in glucose uptake into the heart is made available to cardiomyocytes and stored as glycogen (Fig 1C). Transcriptional analysis on hearts from *Vegfb*^{-/-} mice and controls showed no significant differences in gene expression of rate limiting key enzymes involved in glycogen synthesis or degradation, suggesting that the increase in glycogen most probably arise from increased glucose uptake rather than changes in glycogen synthesis capacity (Fig EV1F). That the incoming glucose is not utilized directly for ATP production in the heart, and instead is stored as glycogen, indicates that glucose is not a limiting factor, or preferred substrate, in cardiac tissue energy consumption in conditions of standard mouse housing.

The impact of increased glucose uptake in settings of VEGF-B inhibition and connection to vascular toxicity is indeed an important aspect. We believe that the capacity of endothelial cells to maintain redox homeostasis is key to withstand toxic injury elicited by hyperglycemia. Importantly, inhibition of VEGF-B in vivo was coupled to an increased rate of glucose uptake leading to increased accumulation in cardiomyocytes and storage as glycogen. We see no signs of that glucose is being stuck in the endothelial cells themselves. Further, as we saw no differential effect on substrate utilization, energy homeostasis or mitochondrial oxidation in response to VEGF-B treatment in the endothelial cells themselves in vitro (Fig EV3), most probably the effect of VEGF-B on endothelial cells regarding glucose is related to transcytosis and not metabolism.

Specific experimental issues:

1. In Fig 1B, how was transcytosis versus glucose metabolism and paracellular movement scored in this transwell assay?

See also response to Reviewer #1, Q1

Trans-endothelial electrical resistance (TEER) measurements were always performed to monitor cellular integrity and exclude that VEGF-B is decreasing paracellular permeability. In addition, VEGF-B does not affect endothelial basal permeability to inulin [3]. Only wells with tight monolayers exhibiting high and equal TEER were subsequently chosen for the experiments. These aspects are described in the Materials and Methods section.

Regarding glucose metabolism, the 2-NBD-glucose tracer cannot be degraded (metabolized) inside the cells, but re-uptake from the lower chamber can occur (further discussed in the response to Reviewer #1, Q1).

2. The GLUT1 imaging looks very non-specific and should be validated using siRNA to GLUT1. The separation techniques used for GLUT1, NRP1 and VEGFR1 are less than optimal since the authors cannot separate ER from PM. This is not surprising since most subcellular techniques for these compartments do not work well in endothelial cells. Also, markers for subcellular compartments are critical for the interpretation of Fig 3C. It is suggested that surface biotinylation of GLUT1 is assessed as is done for LDLR in Fig 6.

See also response to Reviewer #1, Q4

We have now improved the IF staining protocol and updated the panels showing GLUT1 localization in the EC plasma membrane as opposed to intracellular compartments in both Control and VEGF-B stimulated cells (Fig 3A). We have also included quantitative Western blot data showing that the total cellular GLUT1 protein pool was not changed after VEGF-B stimulation up to 2 h (Fig 3B).

A standard method to assess the lipid environment of proteins is by isolation and separation in sucrose density gradients and co-localization with markers for different subcellular compartments. As the reviewer points out, it is also a challenging technique. The subcellular spatial localization of GLUT1 has recently been investigated in HeLa cells using super-resolution imaging [9]. In their paper, GLUT1 distribution in the plasma membrane showed preferred localization to lipid rafts (i.e. cholesterol-rich microdomains) [9]. Intriguingly, treatment with methyl-beta-cyclodextrin (MbCD) reduced clustering of GLUT1 in the lipid raft microdomains [9], in line with our data from MbCD treatment or VEGF-B stimulation leading to reduced free cholesterol loading of plasma membrane, negatively affecting GLUT1 mediated glucose uptake. This reference has now been included in the Discussion in support of our data.

3. The mechanism surrounding VEGFB regulation of plasma membrane cholesterol content and LDLR is interesting, but of some concern. Firstly, the levels of free cholesterol (FC) are measured using the antibiotic filipin that binds FC, not CE. Filipin is notorious for rapidly quenching on slides and tissues, making it extremely difficult for comparative, quantitative measurements. The pictures of the cells are not convincing, and the images are much less convincing in tissues. Based on filipin staining, the VEGF-B treated cells lose 50% of the FC, and this should be either be transported out via ABC transporters or esterified. The one experiment where cholesterol is measured is Fig 4E but this is not cholesterol from the PM, but from the cells. Where is 20% of the cholesterol going since it cannot be metabolized?

In Fig 5, direct measurements of PM cholesterol are needed. An additional control for mbCD would be the uptake of an additional tracer, such as albumin or dextran.

We agree with the Reviewer's concern and are aware of that filipin quenches very easily. In addition to the filipin stain, free cholesterol in cell homogenates was also measured using another method (Fig 4E) to confirm that the free cholesterol pool is reduced in cells treated with VEGF-B. These methods however do not distinguish between plasma membrane and intracellular membrane free cholesterol content. As plasma membrane cholesterol content can be restored by mobilizing

cholesterol from intracellular cholesterol pools, validates however the rationale to assess the total free cholesterol pool [11].

As cholesterol cannot be degraded intracellularly, the cholesterol in endothelial cells is either deposited in membranes, esterified and stored in lipid droplets, transcytosed or excreted out from the cell. Our data showed that the effect of VEGF-B signaling on free cholesterol content in endothelial cells involved the LDLR, but not SR-BI (Scavenger receptor class B type 1), demonstrating some specificity and direction of the mechanism of action. The decreased amount of the LDLR on the endothelial cell surface in response to VEGF-B stimulation was not related to an increased degradation or decreased expression, as both transcript level and total protein content of the LDLR was unaltered in response to VEGF-B stimulation (Appendix Fig S4). This pointed to an effect on endothelial cell LDLR recycling capacity. That the cholesterol is instead esterified and stored in lipid droplets is unlikely, as data from mass spectrometry analysis was showing no compensatory increase in cholesterol esterification in response to VEGF-B signaling in endothelial cells (Fig 4G).

We have in the revised version of the manuscript included a more elaborate and thorough investigation of LDLR expression and LDL uptake in response to VEGF-B stimulation. In Fig 6G, the cell surface expression of the LDLR was assessed by immunostaining with N-terminal-specific antibody on non-permeabilized cells as a complement to the biotinylation experiment in Fig 6F. This confirmed that the LDLR localization at the cell surface is seemingly decreased in response to 10 min of VEGF-B stimulation. This correlated with decreased Dil-LDL binding to the endothelial cells (evaluated at +4 °C to prevent endocytosis) as well as uptake of Dil-LDL (cells incubated in 37 °C) (Fig 6G).

The intracellular LDLR pool was investigated further in Appendix Fig S4C-E and S5. As the perinuclear accumulation of LDLR is less apparent in VEGF-B treated cells (Appendix Fig S4C-D), co-staining with Golgi (Appendix Fig S4E), endosomal (Appendix Fig S5A) and lysosomal (Appendix Fig S5B) markers were analyzed. No apparent changes in the co-localization of the LDLR with these cellular compartments were observed in response to VEGF-B treatment, other than the finding of a more dispersed intracellular distribution pattern in VEGF-B treated cells.

Inhibiting *de novo* cholesterol synthesis in endothelial cells with simvastatin decreased non-esterified cholesterol content as expected and interestingly this also impaired glucose uptake capacity. Importantly, however, simvastatin treatment did not interfere with the effect of VEGF-B, suggesting that cholesterol synthesis and sensing is not involved in the mechanism of action for VEGF-B. Further, we did not detect any transcriptional changes in the *de novo* cholesterol synthesis pathway in hearts from *Vegfb* deficient mice that could account for the increase in non-esterified cholesterol in heart from *Vegfb*^{-/-} mice (Appendix Fig S6).

Collectively, we propose that the observed effect of decreased free cholesterol content in endothelial cells after VEGF-B stimulation is related to impaired LDLR mediated cholesterol uptake.

Regarding the comment about additional controls for MbCD we are not sure if we understand the Reviewer's intention. Uptake of albumin or dextran would act as markers for endocytosis/transcytosis which is potentially interesting, but may not help to fortify the proposed mechanism of action at this point.

4. It is interesting that knockdown of LDLR reduces VEGFB mediated reductions in glucose uptake. Western blotting for LDLR is needed in the KD and VEGFB treated cells. Since LDLR KD attenuates VEGFB-mediated decrease in cholesterol content and glucose uptake, yet siLDLR alone did not significantly decrease cholesterol/uptake by itself. If the mechanism is via recycling of the LDLR, then the si LDLR treated group should be shifted down and not be affected by the VEGFB inhibitor. Controls are needed for LDLR IF, since these Ab are notorious for non-specificity.

We fully agree with the reviewer regarding the issue with LDLR antibody specificities and we have taken measures as to the choice of antibodies (described in Appendix Table S1) and the use of immunoprecipitation technique instead of direct Western blotting.

LDLR protein levels has been evaluated after VEGF-B stimulation by quantitative Western blotting (Appendix Fig S4) and no changes in total protein content was detected.

LDLR expression after LDLR knock-down was evaluated at the mRNA level, approximately 50% reduced, and likely less than that at the protein level, meaning that residual LDLR protein exist in the knock-down experiments which can be enough to ensure proper basal cholesterol homeostasis. In the presence of VEGF-B however, impaired recycling of residual LDLR to the cell surface will lead to a detectable decrease in cholesterol content in the knock-down experiments (Fig 6B). This is different from the *Ldlr* knock-out cells where there is no residual LDLR present to be recycled.

5. In 6D, VEGFB suppression of glucose uptake and the levels of NRP and FLT and another sterol responsive gene (HMGCoaR) are critical for interpretation of the experiments. Again, chemical measurements of FC and CE would strengthen this data.

See responses to Q4 above and Reviewer #1, Q7-8.

We utilized the *Ldlr*^{-/-} mice primarily as a tool to obtain primary endothelial cells lacking the LDLR in order to address and verify the effect of acute VEGF-B signaling on cholesterol and glucose uptake. A difference between knock-out and knock-down of the LDLR is that in the knock-out there is no residual LDLR present.

6. There are no control plasma membrane markers in Fig 6F, making this impossible to interpret.

In updated Fig 6G, the cell surface expression of the LDLR was assessed by immunostaining with N-terminal-specific antibodies on non-permeabilized cells as a complement to the biotinylation experiment in Fig 6F. This confirmed that the LDLR localization at the cell surface is seemingly decreased in response to 10 min of VEGF-B stimulation. This correlated with decreased Dil-LDL binding to the endothelial cells (evaluated at +4 °C to prevent endocytosis) as well as uptake of Dil-LDL (cells incubated in 37 °C) (Fig 6G).

7. If VEGF-B impacts cholesterol, then ratiometric imaging of membrane fluidity should be assessed.

Future studies involving high-resolution imaging are warranted to investigate this aspect in greater detail.

References

- Muhl L, Moessinger C, Adzemovic MZ, Dijkstra MH, Nilsson I, Zeitelhofer M, Hagberg CE, Huusko J, Falkevall A, Yla-Herttuala S, et al. (2016) Expression of vascular endothelial growth factor (VEGF)-B and its receptor (VEGFR1) in murine heart, lung and kidney. *Cell and tissue research*, 10.1007/s00441-016-2377-y
- Mehlem A, Palombo I, Wang X, Hagberg CE, Eriksson U, Falkevall A (2016) PGC-1alpha coordinates mitochondrial respiratory capacity and muscular fatty acid uptake via regulation of VEGF-B. *Diabetes*, 10.2337/db15-1231
- Hagberg CE, Falkevall A, Wang X, Larsson E, Huusko J, Nilsson I, van Meeteren LA, Samen E, Lu L, Vanwildemeersch M, et al. (2010) Vascular endothelial growth factor B controls endothelial fatty acid uptake. *Nature* 464: 917-921
- Hagberg CE, Mehlem A, Falkevall A, Muhl L, Fam BC, Ortsater H, Scotney P, Nyqvist D, Samen E, Lu L, et al. (2012) Targeting VEGF-B as a novel treatment for insulin resistance and type 2 diabetes. *Nature* 490: 426-430
- Makinen T, Olofsson B, Karpanen T, Hellman U, Soker S, Klagsbrun M, Eriksson U, Alitalo K (1999) Differential binding of vascular endothelial growth factor B splice and proteolytic isoforms to neuropilin-1. *J Biol Chem* 274: 21217-21222
- Muhl L, Moessinger C, Adzemovic MZ, Dijkstra MH, Nilsson I, Zeitelhofer M, Hagberg CE, Huusko J, Falkevall A, Yla-Herttuala S, et al. (2016) Expression of vascular endothelial growth factor (VEGF)-B and its receptor (VEGFR1) in murine heart, lung and kidney. *Cell and tissue research* 365: 51-63
- Gaultier A, Simon G, Niessen S, Dix M, Takimoto S, Cravatt BF, 3rd, Gonias SL (2010) LDL receptor-related protein 1 regulates the abundance of diverse cell-signaling proteins in the plasma membrane proteome. *J Proteome Res* 9: 6689-6695

8. Pike LJ (2003) Lipid rafts: bringing order to chaos. *J Lipid Res* 44: 655-667
9. Yan Q, Lu Y, Zhou L, Chen J, Xu H, Cai M, Shi Y, Jiang J, Xiong W, Gao J, et al. (2018) Mechanistic insights into GLUT1 activation and clustering revealed by super-resolution imaging. *Proc Natl Acad Sci U S A* 115: 7033-7038
10. Randle PJ, Garland PB, Hales CN, Newsholme EA (1963) The glucose fatty-acid cycle. Its role in insulin sensitivity and the metabolic disturbances of diabetes mellitus. *Lancet* 1: 785-789
11. Mahammad S, Parmryd I (2008) Cholesterol homeostasis in T cells. Methyl-beta-cyclodextrin treatment results in equal loss of cholesterol from Triton X-100 soluble and insoluble fractions. *Biochimica et biophysica acta* 1778: 1251-1258

2nd Editorial Decision

3 April 2020

Thank you for submitting your revised manuscript. It has now been seen by all of the original referees.

As you can see, the referees find that the study is significantly improved during revision and recommend publication. Before I can accept the manuscript, I need you to address some minor points below:

REFEREE REPORTS

Referee #1:

The authors have addressed all my previous comments and I have no further questions. I think this is an interesting and well-conducted study that will be of high interest to the readership of the journal.

Referee #2:

The authors have addressed my concerns.

Referee #3:

The authors have addressed my comments

2nd Revision - authors' response

7 April 2020

Thank you for the positive response. We have now revised the manuscript and added all the details in the uploaded version.

Corresponding Author Name: Ulf Eriksson

Manuscript Number: EMBOR-2019-49343